# Changes in the Proteomic Profile After Audiogenic Kindling in the Inferior Colliculus of the GASH/Sal Model of Epilepsy

**DOI:** 10.3390/ijms26052331

**Published:** 2025-03-05

**Authors:** Laura Zeballos, Carlos García-Peral, Martín M. Ledesma, Jerónimo Auzmendi, Alberto Lazarowski, Dolores E. López

**Affiliations:** 1Instituto de Neurociencias de Castilla y León (INCYL), Universidad de Salamanca, 37007 Salamanca, Spain; laurazf@usal.es (L.Z.); carlosp@usal.es (C.G.-P.); 2Instituto de Investigación Biomédica de Salamanca (IBSAL), 37007 Salamanca, Spain; 3Departamento de Biología Celular y Patología, Facultad de Medicina, Universidad de Salamanca, 37007 Salamanca, Spain; 4Unidad de Conocimiento Traslacional, Hospital de Alta Complejidad del Bicentenario Esteban Echeverría, Monte Grande B1842, Argentina; mmledesma88@gmail.com; 5Hospital de Alta Complejidad en Red El Cruce Dr. N. C. Kirchner SAMIC, Florencio Varela B1888, Argentina; 6Instituto de Fisiopatología y Bioquímica Clínica (INFIBIOC), Facultad de Farmacia y Bioquímica, Universidad de Buenos Aires, Buenos Aires C1417, Argentina; jeronimo.auzmendi@gmail.com (J.A.); alazarowski@gmail.com (A.L.); 7Consejo Nacional de Investigaciones Científicas y Tecnológicas (CONICET), Godoy Cruz M2290, Argentina

**Keywords:** audiogenic kindling, bioinformatic approaches, epileptogenic nucleus, immune response, proteomic, refractory epilepsy, seizure susceptibility

## Abstract

Epilepsy is a multifaceted neurological disorder characterized by recurrent seizures and associated with molecular and immune alterations in key brain regions. The GASH/Sal (Genetic Audiogenic Seizure Hamster, Salamanca), a genetic model for audiogenic epilepsy, provides a powerful tool to study seizure mechanisms and resistance in predisposed individuals. This study investigates the proteomic and immune responses triggered by audiogenic kindling in the inferior colliculus, comparing non-responder animals exhibiting reduced seizure severity following repeated stimulation versus GASH/Sal naïve hamsters. To assess auditory pathway functionality, Auditory Brainstem Responses (ABRs) were recorded, revealing reduced neuronal activity in the auditory nerve of non-responders, while central auditory processing remained unaffected. Cytokine profiling demonstrated increased levels of proinflammatory markers, including IL-1 alpha (Interleukin-1 alpha), IL-10 (Interleukin-10), and TGF-beta (Transforming Growth Factor beta), alongside decreased IGF-1 (Insulin-like Growth Factor 1) levels, highlighting systemic inflammation and its interplay with neuroprotection. Building on these findings, a proteomic analysis identified 159 differentially expressed proteins (DEPs). Additionally, bioinformatic approaches, including Gene Set Enrichment Analysis (GSEA) and Weighted Gene Co-expression Network Analysis (WGCNA), revealed disrupted pathways related to metabolic and inflammatory epileptic processes and a module potentially linked to a rise in the threshold of seizures, respectively. Differentially expressed genes, identified through bioinformatic and statistical analyses, were validated by RT-qPCR. This confirmed the upregulation of six genes (*Gpc1*—Glypican-1; *Sdc3*—Syndecan-3; *Vgf*—Nerve Growth Factor Inducible; *Cpne5*—Copine 5; *Agap2*—Arf-GAP with GTPase domain, ANK repeat, and PH domain-containing protein 2; and *Dpp8*—Dipeptidyl Peptidase 8) and the downregulation of two (*Ralb*—RAS-like proto-oncogene B—and *S100b*—S100 calcium-binding protein B), aligning with reduced seizure severity. This study may uncover key proteomic and immune mechanisms underlying seizure susceptibility, providing possible novel therapeutic targets for refractory epilepsy.

## 1. Introduction

Audiogenic kindling is a model used in epilepsy research, particularly to study genetic predispositions and the brain’s response to repeated auditory stimuli that can trigger seizures [1]. This model has been extensively applied in rodents that are genetically prone to audiogenic seizures, such as Wistar Audiogenic Rats (WARs) [2] and Krushinsky–Molodkina (KM) rats [3], among others. This gradual change mirrors the mechanisms involved in human epilepsy, particularly in the development of generalized tonic–clonic seizures (GTCS) [4,5].

During this process of repeated auditory stimuli, there is a progressive enhancement of seizure activity, which involves plastic reorganizations of synaptic activity [6,7], neurochemical modifications in the affected areas [8], and the recruitment of broader regions of the brain, including limbic structures like the amygdala [2,9]. Among all the changes that occur, the proteins associated with the mechanisms of audiogenic kindling are important, since it is known that in audiogenic epilepsy the proteins may be involved in synaptic function [10], neurotransmitter transporters [11], neuronal excitability [8,12], and signal transduction [13], which are all processes that are modified during audiogenic kindling [14,15].

Using the GASH/Sal (Genetic Audiogenic Seizure Hamster, Salamanca) audiogenic epilepsy model [16] under basal conditions (GASH/Sal naïve) and after an audiogenic kindling process (GASH/Sal sAUK), we studied the protein changes in the epileptogenic nucleus to elucidate the role of certain proteins in the maintenance of seizures. We also confirmed these data with gene expression studies.

Understanding the protein changes associated with the process of repeated seizures, a phenomenon that occurs in refractory epilepsies [17], offers the possibility of modulating these proteins and thus could lead to better seizure control with fewer side effects. Proteomics has the potential to transform epilepsy research by offering new avenues for diagnosis and treatment, bringing us closer to more effective therapies and personalized medicine in epilepsy management.

## 2. Results

Naïve and stimulated animals were submitted to different approaches to elucidate behavioral and proteomic differences between the two groups. Interleukin and Auditory Brainstem Response (ABR) analyses were performed in naïve and stimulated hamsters once the protocol was ended. Moreover, the seizure severity index was described, and different bioinformatic approaches were fulfilled to study proteomic differences in the epileptogenic foci (inferior colliculus) of naïve and stimulated GASH/Sal hamsters.

### 2.1. Severity Index

GASH/Sal animals were exposed three times per day for fifteen days to acoustic stimuli. For each stimulus, we analyzed the mean severity index of all hamsters using a modified scale adapted to audiogenic seizures [18]. As observed in Figure 1, the seizure severity gradually decreased along the protocol. Thus, we decided to compare the severity indexes among all animals to determine if they could be differentiated according to their response to sound stimuli along the protocol. Therefore, all stimulated GASH/Sal animals were subjected to Dunn’s multiple comparison test (Appendix A). As a result, we obtained two groups that were subsequently named as “responders” (sAUK.R) and “non-responders” (sAUK.NR). The sAUK.R group corresponded to animals with a higher severity index mean over the course of the protocol, while the other included hamsters with a lower severity index. Each of these two groups consisted of animals with a similar mean severity index that was significantly different from the mean of the other group.

Knowing about the existence of two stimulated hamster groups, we decided to analyze the severity index differences between these two groups for each stimulus (Appendix A). As a result, we detected significant differences regarding the mean severity index between the two groups from the 15th stimulus onward, with exceptions at the 17th and 38th stimulus (Figure 2).

To make sure of the proper functioning of the auditory system in the non-responder GASH/Sal hamsters, we decided to perform the auditory brainstem response test before and after the audiogenic kindling protocol.

### 2.2. Auditory Brainstem Response

Latencies and amplitudes of the five waves, representing the different nuclei involved in the auditory pathway and hearing threshold, were studied with the Auditory Brainstem Response analysis. For the threshold analysis, a slightly non-significant increase in the hearing threshold was detected at the end of the audiogenic kindling compared to the initial point (*p* > 0.05). GASH/Sal hamsters, like other audiogenic genetically prone seizure animals, have pathologically elevated auditory thresholds at naïve conditions [16]. After repetitive sound stimuli, the auditory threshold worsened, increasing from 64.29 ± 2.02 dB up to 70 ± 0.01 dB for the left ear and from 65.71 ± 2.02 dB to 71.43 ± 1.43 dB for the right ear (Figure 3). Nevertheless, the animals reacted correctly to suprathreshold stimuli of 90 dB during the ABR protocol, indicating that they can listen to the high-intensity stimuli of the audiogenic kindling protocol. Moreover, the wave shapes are clear for both ears at the end of the protocol, indicating correct processing of auditory information through the pathway (Figure 4).

Related to the analysis of the amplitude and latency of each wave, we could observe significant changes in the amplitude of wave I in both ears (*p* < 0.05 for both left and right ears). For the left ear, the mean amplitude of wave I at the initial time was 0.3983 ± 0.014 mV, while it decreased to 0.3335 ± 0.02036 mV for the final time. No significant differences were detected regarding the amplitude of the rest of the waves, although it is worth mentioning that the amplitude for some waves was even higher after the protocol than at the start. The analysis for the right ear was similar, as the only significant change detected (*p* < 0.05) was related to the amplitude of wave I, which was higher at the initial time (0.3548 ± 0.02153 mV) than at the final point (0.2881 ± 0.01635 mV). There were no significant changes regarding wave latencies for the left or right ear (Figure 5). The waves’ amplitude values indicate the synchronized neuronal activity for each point of the auditory pathway. These results presumably indicate the existence of a lower number of firing neurons in the auditory nerve (wave I) at the end stimulation time, without a significant decrease of the neuronal activity in the rest of the points of the auditory pathway. No significant differences were observed in the latency between the initial and final time points for any of the waves studied, indicating that the protocol does not affect the processing time following sound stimulation.

### 2.3. Cytokines

Knowing about the important and dual role of the immune system in epileptogenesis, we decided to study the differences in circulating cytokines levels between the non-responder GASH/Sal animals and naïve GASH/Sal animals after they completed the audiogenic protocol. We observed mainly higher levels of most proinflammatory cytokines in the blood of non-responder hamsters compared to the GASH/Sal naïve group. Some of these detected proteins have been related in the literature to seizure generation, such as IL-1 alpha (*p* < 0.01), IL-10 (*p* < 0.01), or TGF-beta (*p* < 0.001). We only observed lower levels of the Insulin-like Growth Factor-1 (IGF-1) in sAUK.NR hamsters compared to non-stimulated animals (Figure 6). Detailed data of cytokine level differences between non-responder sAUK animals and naïve animals are shown in Appendix A.

### 2.4. Proteomics

The inferior colliculus (IC) is responsible for seizure genesis in animals that are genetically prone to audiogenic seizures. Proteomic differences in the epileptogenic foci between non-responder and naïve hamsters could clarify the mechanism responsible for decreased severity in a group of hamsters. For that purpose, we used next-generation label-free quantitative proteomics in data-independent acquisition (DIA) mode to perform differential proteomic analysis of the GASH.sAUK.NR and GASH.naïve groups.

Samples were processed on different days; 43,325 peptides and 5775 proteins were quantified in the first experiment. The missing value (NA = not available value) treatment approach (Section 4.8) eliminated 921 proteins, resulting in 4854 under analysis. In the second experiment, 95,153 peptides and 8434 proteins were quantified. The NA treatment approach eliminated 836 proteins, resulting in 7598 proteins under analysis.

#### 2.4.1. Batch Effect Correction

Both datasets were merged, and only the common 4468 proteins were considered for batch effect correction. The batch effect correction was successfully applied to the final dataset, and the principal component analysis (PCA) showed that the data were no longer clustered by the batch day (Figure 7).

#### 2.4.2. Proteomic Data Analyses for Uncovering Potential Status Epilepticus Regulation in GASH/Sal Model

Different approaches were followed to discover the patterns, biomarkers, or pathways responsible for the disparities between non-responders and naïve hamsters that could explain their differences related to the severity of their convulsions. Those approaches included unsupervised analysis, univariate analysis, gene set enrichment analysis, and weighted correlation network analysis. The most significant and interesting proteins from each approach were selected for further RT-qPCR (Real-Time Quantitative Reverse Transcription PCR) validation.

##### Unsupervised Analysis

First, the ability of the total (4468) detected hamster proteins to separate groups was evaluated using unsupervised analyses, specifically employing the Uniform Manifold Approximation and Projection (UMAP) method. UMAP was mainly used for dimensionality reduction and pattern visualization rather than defining discrete clusters. When analyzing all detected proteins, a clear separation between groups was observed, although samples from different groups remained relatively close in the UMAP space. Specifically, samples from Group 1 exhibited UMAP2 values ranging from 0 to 2 and UMAP1 values from −1 to 1, while samples from Group 2 had UMAP2 values ranging from 0 to −1.5 and UMAP1 values from −1 to 1.5. Despite their proximity, these groups formed distinguishable regions within the UMAP projection (Figure 8).

These proteins, initially identified in *Mesocricetus auratus*, were mapped to *Homo sapiens* gene symbols. The predefined gene set consisting of 4108 Gene Symbols of *Homo sapiens* was then processed to eliminate those hits with isoforms. The criteria only retained the most significant protein isoforms, resulting in a total of 4085 proteins listed in Appendix A.

##### Univariate *t*-Test Analysis

In the comparison between GASH.sAuk.NR and GASH.naïve groups, the univariate analysis identified 159 differentially expressed proteins (DEPs) based on a *p*-value < 0.05 and a Cohen’s effect size (CES) ≥ 2 or ≤−2, respectively (Figure 9, Appendix A). The log of protein abundance was used as the dependent variable in this analysis, as it is the standard measure in proteomic studies for assessing differential expression. FDR-adjusted *p*-values were also calculated to control for multiple comparisons and are provided in Appendix A. Among the 159 proteins identified, 10 were upregulated with CES values ranging from 2 to 3.39, while 149 were downregulated, with CES values ranging from −2 to −8.89. The combination of the *p*-value and CES ensures statistical rigor while preserving biologically relevant findings.

Subsequently, we evaluated the ability of the 159 DEPs to separate the groups using unsupervised analysis. In contrast to Figure 8, the separation became even more pronounced when analyzing only the differentially expressed proteins. Samples from Group 1 displayed UMAP2 values ranging from 7 to 10 and UMAP1 values from 0 to 2, whereas samples from Group 2 exhibited UMAP2 values ranging from −5 to −8 and UMAP1 values from −2 to 0. This enhanced separation suggests that differential protein expression contributes significantly to group distinction (Figure 10).

### 2.5. Gene Validation of the Overexpressed DEPs

In total, 149 underexpressed DEPs and 10 overexpressed DEPs (oDEPs) were detected in the inferior colliculus of GASH.sAUK.NR compared to GASH/Sal naïve hamsters. Overexpressed proteins (CES ≥ 2.0) were selected for further validation. When comparing the expression levels of the protein-coding genes between stimulated and naïve animals, we found higher expressions of several transcripts in the sAUK.NR group: Glypican-1, *Gpc1*, (1.641 ± 0.09; *p* < 0.001); Syndecan-3, *Sdc3*, (1.118 ± 0.07; *p* < 0.01); Nerve Growth Factor Inducible, *Vgf*, (1.887 ± 0.129; *p* < 0.001); Copine 5, *Cpne5*, (1.469 ± 0.109; *p* < 0.001); the Arf-GAP with GTPase domain, ANK repeat, and PH domain-containing protein 2, *Agap2*, (1.377 ± 0.154; *p* < 0.05); and Dipeptidyl Peptidase 8, *Dpp8*, (1.189 ± 0.054; *p* < 0.05). No significant differences were detected for the expression of Carboxypeptidase E (*Cpe*), GTPase Activating Protein (SH3 Domain) Binding Protein 2 (*G3bp2*), the MAP Kinase Activating Death Domain (*Madd*), or the Inhibitor of kappaB Kinase Gamma (*Ikbkg*). Thus, six differentially overexpressed proteins were confirmed by the validation of their protein-coding genes by RT-qPCR (Figure 11).

### 2.6. Gene Set Enrichment Analysis

To further investigate the underlying biological processes, we performed Gene Set Enrichment Analysis (GSEA) to identify enriched pathways associated with non-responders and naïve hamster groups.

The input to GSEA comprised 4085 gene symbols with associated rank values. GSEA output reported no significant upregulated curated pathways in GASH.sAUK.NR compared to GASH.naïve (Appendix A); however, the algorithm found 14 significant downregulated curated pathways in the same comparison (Appendix A). Many significant pathways were either related (parent and child terms) or redundant across databases, as illustrated in Figure 12A. Furthermore, the peak frequency of Cohen’s effect sizes for proteins in these pathways was located around −1.75 (Figure 12B). Subsequently, the Enrichmentmap tool [19] grouped these pathways in two main biological themes. The most significant terms of each cluster were the following: “Electron transport chain oxphos system in mitochondria” and “Glycolysis and gluconeogenesis”. Cluster membership of all enriched terms is indicated in the last column of Appendix A. In the “Glycolysis and Gluconeogenesis” category, a directly related pathway with epilepsy was found: “Metabolic epileptic disorders”. This pathway was analyzed with PathVisio [20] for searching key genes for further analysis. An image merging Enrichmentmap and PathVisio visualization can be observed in Appendix A. According to the criteria established in the methodology, we looked for genes encoding proteins involved in significant epilepsy-related pathways. Genes with the highest enrichment score (ES) values in the significant “metabolic epileptic disorders” pathway were selected for validation: *Slc25a1* (Solute Carrier Family 25 Member 1), *Shmt2* (Serine Hydroxymethyltransferase 2), and *Dld* (Dihydrolipoamide Dehydrogenase). Additionally, chemokines have been shown to play a role in the inflammatory pathogenic of epilepsy [21]; therefore, the top three genes with the highest ES values relative to the significant “C-X-C chemokine receptor type 4 (CXCR4)” pathway were selected: the RAS-like proto-oncogene B (*Ralb*), Itchy E3 Ubiquitin Protein Ligase (*Itch*), and FYN Proto-Oncogene of the Src Family Tyrosine Kinase (*Fyn*).

### 2.7. Gene Validation of GSEA-Selected Proteins

Concerning the three selected genes of the “Metabolic epileptic disorders” pathway (*Slc25a1*, *Shmt2*, and *Dld*), we did not detect any significant difference in the epileptogenic foci between sAUK.NR GASH/Sal and naïve hamsters (Figure 13A). On the other hand, among the selected genes of the CXCR4 pathway for validation, we observed a significantly lower *Ralb* expression in the GASH.sAUK.NR group compared to the GASH.naïve hamsters (0.7949 ± 0.054; *p* < 0.05). No significant differences were detected between the experimental groups for the remaining genes, *Itch* and *Fyn* (Figure 13B).

### 2.8. Network Analysis

Significant protein clusters potentially associated with epilepsy were identified following the application of the WGCNA (Weighted Gene Co-expression Network Analysis) algorithm [25,26]. A weighted protein co-expression matrix was constructed, including the 4085 human orthologous genes, as described in Section 2.4.2. (Unsupervised Analysis). We found 17 modules, with one (Module green) showing a significant association with the epilepsy status variable (*p*-value < 0.05) (Figure 14D). Module green included 186 significant proteins (*p*-value < 0.05), 108 of which were also significant in the univariate *t*-test (*p*-value < 0.05 and CES ≤ −2 or ≥2). A full table providing gene module membership (MM) and protein significance (PS) within the epilepsy group variable is provided in Appendix A. Finally, significant proteins in the WGCNA were submitted to the String Database (version 12.0) for uncovering protein–protein interaction networks [27]. We found 11 clusters applying the k-means clustering method with a high cut-off value (0.7) for the interaction confidence score (Appendix A). The first cluster included a high-density node network of 74 proteins (39.78% of the total significant proteins in Module green). The top five proteins with the highest epilepsy-related scores (combined scores from Disease 2.0 [28] and Disgenet [29] databases) were screened for RT-qPCR validation (Table 1). A full table containing the epilepsy-related scores for 74 proteins from cluster 1 is provided in the Appendix A. Finally, a comprehensive visualization of the distribution of epilepsy-related scores in this cluster is also available in Appendix A.

### 2.9. Gene Validation of WGCNA-Selected Candidates

Five candidates from *WGCNA* were selected for molecular validation: *Slc1a2*, *Slc1a3*, *Akt1*, *Cyfip2*, and *S100b*, with all of them related to epilepsy according to the aforementioned disease databases. Of those five genes, contradictory results were obtained for the *Slc1a2* and *Akt1* genes (Figure 15), as we detected higher significant expression levels in the IC of the GASH.sAUK.NR group compared to the GASH.naïve group (1.189 ± 0.054, *p* < 0.05, and 1.115 ± 0.034, *p* < 0.01, respectively). On the contrary, lesser expression of the *S100b* gene was detected in the IC of the stimulated group compared with the naïve hamsters (0.838 ± 0.042; *p* < 0.05). We did not observe significant changes for the *Slc1a3* and *Cyfip2* genes.

## 3. Discussion

In this work, we show that GASH/Sal hamsters prone to audiogenic seizures [16], when exposed to a protocol of repeated stimulations (audiogenic kindling) [1,2,4], experience a marked decrease in the severity score of the seizures as the number of applied stimuli increases. Based on this, two populations of GASH/Sal hamsters with different dynamics of decay of the response to the sound stimulus were observed. On the one hand, we found a population in which the decay to the response was noticeable after 30 stimuli, characterized by a high rate of variability. We consider that these hamsters remained reactive to the stimulus; therefore, we named them responders to the audiogenic kindling stimulus. The high variability rate could indicate that this GASH/Sal population is composed of several subpopulations with different decay rates in the seizure severity score. However, we did not explore this possibility because the second population we observed showed a very marked decrease in response to the stimulus that was evident from the fifteen stimuli onwards. Strikingly, the decay in response to the sound stimulus had little variability and decreased to the lowest values of the seizure severity scale. Some animals remained seizure-free. For this reason, we consider that this group of hamsters stopped responding to the stimulus, and we named them non-responders to the audiogenic kindling stimulus (sAUK.NR).

In audiogenic seizure-prone animals, the epileptogenic focus is the inferior colliculus, and the mesencephalic structures involved belong to the auditory circuit [30].

As audiogenic kindling progresses, mesencephalic seizures change to limbic seizures, with a focus mainly on the amygdala [2]. This could suggest that the decrease in the severity of the observed seizures could be due to a lack of recruitment of the limbic structures or to deficiencies in the auditory pathway. The classic literature on the kindling protocol suggests an increase in the severity of epileptic seizures [31,32]. However, our model exhibited a reduction in seizure severity, accompanied by a decrease in wave I of the ABR and metabolic downregulation, among other significant changes. These observations can be explained by model-specific mechanisms and recent findings in the high-impact literature, which highlight the variability in responses to kindling depending on genetic, metabolic, and neurophysiological contexts. Recent studies have demonstrated that the response to kindling can vary significantly depending on the genetic background and the type of experimental model. For example, Pitkänen et al. emphasize that genetic models of epilepsy may exhibit unique compensatory mechanisms that modulate neuronal excitability and seizure progression. These mechanisms may include the regulation of ion channels, synaptic plasticity, and the modulation of specific neural networks, which could explain the reduction in seizure severity observed in our model [33]. Additionally, the metabolic downregulation observed in our model might be associated with neuroprotective mechanisms that reduce neuronal excitability. Lennox et al. reported that reduced metabolic activity in certain epilepsy models may be linked to decreased production of reactive oxygen species and improved energy efficiency, which prevents neuronal hyperexcitability [34]. This supports our hypothesis that the metabolic downregulation found in our model could contribute to the reduction in seizure severity.

Brainstem status investigation is a neurological test that provides information on the auditory threshold, the number of neurons with synchronous activity (amplitude), and the speed of signal transmission (latency) [35]. We compared the brainstem-evoked responses of GASH/Sal hamsters before the start of the protocol and after the end of audiogenic kindling. We only observed a significant decrease in the amplitude of wave I, reflecting the synchronous activity of the auditory nerve neurons. Notwithstanding, we did not detect significant changes in the amplitudes of the rest of the waves or in any of the latencies. As pointed out by several studies, this could indicate the existence of compensatory mechanisms by the central structures of the auditory system to counteract the decrease in the activity of the auditory nerve neurons [36,37,38]. The mechanism by which this balance of the pathway occurs is unknown, but it could be due to either an increase in excitatory synchronous activity in the rest of the structures or to a loss of inhibitory synapses [39]. We have not investigated the biological possibilities causing the maintenance of the rest of the waves’ amplitudes of the ABRs in the GASH/Sal hamsters after the kindling protocol. Compensation at the central level is probably due to the action of several of the mechanisms mentioned before which, as a whole, end up having a compensatory effect on signal transduction. In our animal model, we could also point out to a worsening of the cochlear characteristics in the GASH/Sal hamsters. In fact, data from our laboratory evidenced the existence of aberrant morphofunctional features of the cochlear nucleus in naïve GASH/Sal hamsters. Also, there were several expression changes in the same foci. We proved a decreased expression of *Vglut1* and *Vglut2* transcripts in the cochlea of naïve GASH/Sal hamsters. VGLUT1 and VGLUT2 are glutamate transporters indispensable for the excitatory synapsis between the auditory nerve fibers and the cochlear nucleus. The audiogenic kindling protocol could worsen this condition, as cochlear neuropathy is influenced by noise exposure. As a consequence, there would be decreased neuronal activity at that level, which correlates with the aberrant decreased in wave I amplitude after the sAUK protocol. Furthermore, constant auditory stimuli could also affect the expression of glutamatergic transporters such as VGLUT1 and VGLUT2, modifying the synaptic vesicle composition [38]. A deep insight into the expression changes, mainly in the inferior colliculus, would be helpful to clarify the physiology under the compensation in the epileptogenic foci.

Therefore, non-responding GASH/Sal hamsters only showed a decrease in the synaptic activity of auditory nerve neurons, without changes in the rest of the pathway. This implies a good functioning at the level of the central nervous system in response to auditory stimuli. In addition, after the protocol, they did not show a significant worsening of their auditory threshold.

In order to elucidate the mechanisms by which a set of animals undergoes seizures of lower severity, we decided to analyze the proteomic profile of the epileptogenic focus of non-responding hamsters and compare it with animals in naïve conditions, since the differences in the response to the sound stimulus became statistically evident from the fifteenth stimulus onwards. In addition, we decided to analyze the circulating cytokine plasma profile, since inflammatory processes play an important role in epileptogenesis [40,41].

### 3.1. Cytokines

Immunoblot analysis revealed differences in the blood cytokine levels between sAUK.NR and naïve hamsters. Although sAUK.NR hamsters had a decreased response to stimulation, high plasma levels of proinflammatory cytokines were found after treatment, indicating a generalized inflammatory state. Inflammatory cytokines, such as IL-1 beta, TNF-alpha, and IL-6, which were increased by 2–4-fold, can infiltrate the brain but can also be secreted by activated microglia in the central nervous system (CNS), promoting the observed reactive gliosis of epileptic brains [42,43] and even disrupting the blood–brain barrier (BBB) [44], while loss-of-function experiments prevent epileptogenesis [45,46,47]. Furthermore, elevated plasma levels of other inflammatory cytokines have been linked to epilepsy. IFN-gamma concentration has been correlated with severe seizures [48], MIP-1 alpha was found to be at an increased concentration in the serum of patients with intractable temporal lobe epilepsy [49], and increased TARC (a TARC/sICAM5 ratio) was detected in the plasma of focal epileptic patients [50]. Moreover, elevated IL-1 alpha has been found in the brains of patients after seizures [51]. In line with these results, there was a 50% decrease in Insulin-like Growth Factor 1 (IGF-1) plasma levels in the sAUK.NR group, since IGF-1 can inhibit the secretion of proinflammatory cytokines such as IL-1 beta, IL-6, and TNF-alpha [52]. Low serum concentrations of IGF-1 have been related to more prolonged and severe cases of epilepsy [53]. In addition, IGF-1 promotes hippocampal neurogenesis in epilepsy [54]. However, excessive neurogenesis can increase the number of excitatory terminals and unbalance the excitation/inhibition ratio [54], so its plasma decrease may be beneficial by decreasing spurious synaptogenesis. In contrast to this pro-inflammatory picture, a 3- to 4-fold increase in IL-10 was observed. Strikingly, a high increase (4- to 5-fold) in the Granulocyte Colony Stimulating Factor (GCSF) plasma concentration was found. Although the GCSF has a well-recognized role in the maturation of neutrophils in the bone marrow, it can also produce the stimulation of nuclear polymorphs in the circulation, complicating the inflammatory landscape. However, at the level of the CNS, it has an antiapoptotic and neuroprotective effect and increases neuroplasticity [55,56]. Even more striking is the increase in Fractalkine, which has recently been described with antiepileptic properties and decreased in the serum of patients with drug-resistant epilepsies compared to patients with controlled epilepsies and healthy individuals [57]. Together, these results suggest a fine-tuning mechanism between pro/anti-inflammatory conditions that could result in the loss of sensitivity to audiogenic kindling.

### 3.2. Proteomics

GASH/Sal hamsters are genetically predisposed to seizures in response to high-intensity auditory stimuli. However, other mechanisms appear to inhibit the generation of more severe seizures and may affect this genetic trait. For this reason, we decided to analyze the proteomic profile of the epileptogenic foci of sAUK.NR hamsters versus animals in naïve conditions. The fundamental objective would be the identification of biomarkers and/or pathways linked with the rise in the threshold of seizures.

The analysis of the proteomic profile revealed 159 differentially expressed proteins out of a total of 4085 proteins, of which 10 were overexpressed (oDEPs) and 149 were underexpressed (uDEPs). Among the oDEPs are proteins associated with cell survival functions, although only some of them have been directly linked to epilepsy. In this sense, the inhibitor of kappaB kinase gamma (IKBKG) controls the activity of the transcription factor NF-κB, which regulates the expression of hundreds of genes in almost all cells and is involved in cell proliferation, cell survival, response to cellular stress, innate immunity, and inflammation [58]. IKBKG produces an upregulation of NF-κB, a transcription factor that plays a critical role in epilepsy, since under normal conditions the transcription factor is retained in the cytoplasm by its inhibitor. At the same time, its phosphorylation uncouples the inhibitor-NF-κB complex and allows the entry of the transcription factor into the nucleus. Some studies have suggested that NF-κB upregulation plays a short-term neuroprotective role after seizures in animal models [59], which may be due to an increase in the expression of proteins associated with neuronal survival [60]. Recent studies in primary neuronal cultures showed that increased extracellular glutamate concentration promotes nuclear translocation of NF-κB and the consequent increase in the expression of survival-related markers such as ABCB1 and EPOR [40]. sAUK.NRs also overexpressed Madd, the MAPK-activating protein containing a death domain (or MAP kinase-activating death domain) implicated in, among others, neurotransmission (Rab3 GEFs and effectors playing a role in synaptic vesicle formation/trafficking) and cell survival upon TNF-alpha treatment [61]. *MADD* positively regulates a downstream step of synaptic exocytosis [62], and mutations in this gene have been included in the early-onset or syndromic epilepsy v4.164 panel [63].

In the context of neuronal survival stimulation, sAUK.NR. also overexpressed Vgf (Nerve Growth Factor Inducible). The main neurons expressing VGF are the excitatory and glutamatergic ones of the hindbrain and hypothalamus, whereas glial single cells are insignificant [64]. The overexpression of murine Vgf in mixed primary cultures of spinal cord neurons protected against excitotoxic injury induced by AMPA and NMDA glutamate receptor agonists [65]. This might attenuate excitotoxic injury in the spinal cord of patients with amyotrophic lateral sclerosis and other neurodegenerative diseases. Furthermore, neuroendocrine regulatory peptides 1 and 2, derived from the neurosecretory protein VGF, inhibit the excitability of magnocellular neurosecretory cells in the hypothalamus [66], activating inhibitory GABAergic interneurons. This mechanism may explain the rise in the threshold of seizures in this group of non-responding animals.

Another overexpressed protein was Syndecan-3 (Sdc3), a transmembrane heparan sulfate proteoglycan of the CNS. Loss of the *Sdc* gene results in seizure phenotypes in *Drosophila* [67]. Alternatively, the loss of Sdc in postmitotic neurons could alter cell survival and induce neuron loss, a fact that has been linked to synchronization and tonic depolarization leading to seizures [68]. Syndecan can be downregulated by IGF2BP (Insulin-like Growth Factor 2 (IGF-II) mRNA Binding Protein), and increased IGF2BP expression is linked to seizure occurrence [69].

Glypican 1 (Gpc1), a glycoprotein widely expressed in the developing and adult CNS, was also found to be overexpressed in the IC of the sAKU.NRs and plays a role in regulating multiple signaling pathways [70]. In addition to its role in tumorigenesis, GPC1 is also involved in brain development [71], neurodegeneration and neurogenesis [72], axonal guidance, and regeneration [73]. Although no clear relationship between GPC1 and epilepsy has been described, another member of the same proteoglycan family, GCP4, has been reported to promote mossy fiber sprouting after pilocarpine-induced status epilepticus via mTOR [74]. Additionally, the overexpression of GCP4 was detected in epileptic patients, and its downregulation decreased seizure frequency in experimental models [75].

Another protein indirectly linked to epilepsy is Copine 5, a calcium-dependent phospholipid binding protein. There is direct evidence that epileptic seizures induce CPNE6 expression in both rats and humans [76]. However, nothing has been described about CPNE5 concerning epilepsy, although lower CPNE5 levels are significantly associated with decreased survival rates [77]. Furthermore, multiple myeloma patients with higher CPNE5 expressions reported longer event-free survival and overall survival [78].

In this context, there is also AGAP2, a GTPase/GTP activating protein involved in the actin remodeling and receptor recycling system [79], whose overexpression inhibits apoptosis [80]. Interestingly, TGFβ1 has been reported to be a positive regulator of AGAP2 expression [81], which is consistent with the elevated levels of TGFβ found in the sAUK.NRs. Concerning elevated proinflammatory cytokines, the enzyme Cpe (carboxypeptidase E) that produces the mature form of enkephalin was also overexpressed. It is now recognized that this enzyme plays a general role in producing neuropeptides and peptide hormones [82]. In addition to its peptidase activity, CPE is a neurotrophic factor that promotes neuronal survival, recently renamed Neurotrophic Factor-α1 (NF-α1) [83,84]. This protein has been reported to protect neurons from stress-induced cell death in the hippocampal CA3 region of mice [85]. Hippocampal delivery of the *carboxypeptidase E* gene has been shown to prevent neurodegeneration in male mice with Alzheimer’s disease [86], while *Cpe* knockout mice subjected to stress showed a loss of hippocampal CA3 neurons similar to that produced by status epilepticus [87].

Dipeptidyl peptidase 8 (DPP8) is an intracellular protein that has been postulated to have a role in regulating apoptosis, proliferation, and interaction with the extracellular matrix immune response [88]. Somehow, DPP8 is related to cell survival, as DPP8 inhibitors induce apoptosis in some types of cancer [89]. Furthermore, it protects against autoinflammatory diseases [90], suppressing the spontaneous activation of the protein that forms inflammasome [91].

Finally, although not directly related to epilepsy, G3BP2 (GTPAse Activating Protein (SH3 Domain) Binding Protein 2) belongs to a family of RNA-binding proteins and regulates the nucleoplasmic shuttle, thus participating in a variety of biological functions such as cell growth, differentiation and migration, and RNA and protein metabolism [92]. G3BP2 is essential for the assembly of stress granules (SGs), which provide a way for eukaryotic cells to respond to various sources of stress, such as oxidative conditions, heat shock, ultraviolet light, and viral infection, by forming cytoplasmic components [93]. SGs are involved in a variety of biological functions, including the response to apoptosis and inflammation. Recently, a new type of cell death, characterized by imbalances in iron metabolism and excessive generation of reactive oxygen species—known as ferroptosis—has been linked to epilepsy [94]. The overexpression of G3BP2 can initiate the assembly of SGs; in this case, precipitates of ferritin loaded with excess iron (typical of ferroptosis [95]) provide survival advantages in an unfavorable environment [96].

Excitotoxicity, ROS production, and apoptosis cause the initial neuronal damage in epilepsy. The presence of these proteins is in some way facilitating resistance to seizures, either because of their neuroprotective effect (Cpe, Cpne5, Vgf, G3bp2, Gpc1, Ikbkg, and Sdc3) or because they inhibit the apoptotic processes associated with the development of seizures (Agap2, Dpp8, and Madd). Also, one of the overexpressed proteins (Sdc3) is linked to the onset of seizures, and its reduction triggers seizures, so high levels of this protein would exert a braking effect on the onset of seizures. In addition, some of these proteins directly affect the levels of inhibitory and/or activating neurotransmitters. Moreover, one of the most expressed proteins, Vgf, inhibits glutamatergic stimulation and activates GABAergic stimulation, which would be in accordance with greater resistance to seizure generation.

#### 3.2.1. GSEA

Gene Set Enrichment Analysis suggested two downregulated pathways directed linked to epilepsy. On the one hand, the epileptic metabolic disorders pathway whose candidate genes for validation were *Dld*, *Slc25a1A*, and *Shmt2*. In addition to being linked to patients with hepatic and metabolic disorders, DLD deficiency has been associated with a range of diseases exhibiting a broad phenotypic spectrum, including neurological issues such as ataxia, seizures, and intellectual disability [97]. This is because *DLD* encodes an enzyme involved in glycine metabolism that functions as a mitochondrial E3 ubiquitinase [98]. In parallel, mutations in the mitochondrial citrate transporter gene *SLC25A1* have been linked to neurometabolic disorders characterized by mitochondrial citrate excretion deficiency, impairing lipid, cholesterol, and sphingolipid synthesis, which are essential for proper neuronal function. This leads to a range of pathological phenotypes, including neonatal encephalopathies, psychomotor development issues, resistant seizures, and respiratory problems [99]. Related to glycine and folate metabolism, deficiencies in the expression level or activity of the SHMT2 enzyme led to decreased glycine levels, loss of mitochondrial homeostasis and folate deficiency, which together lead to microcephaly, intellectual disability, and cardiomyopathies [100].

Contrary to expectations, RT-qPCR validation did not indicate significant differences, suggesting that the expression level of these proteins could have a post-translational regulation step independent of the mRNA level.

On the other hand, the second pathway involved was related to inflammation mediated by the CXCR4 receptor. First, *Itch* was analyzed, it acts as an E3 ubiquitin ligase of the HECT domain. This enzyme marks some proteins for lysosomal degradation, such as CXCR4 [101]. It also inhibits inflammation by marking proinflammatory transcription factors for degradation (AP-1 and JunB), regulates the expression of IL-1 alpha, and controls the differentiation of helper T cells 2 (Th2) [102]. In addition, its deficiency promotes antigen-driven B-cell responses [103]. We also analyzed the expression of *Fyn*, since it is a member of the Src family of tyrosine kinases that participates in several biological processes such as cell growth, survival, adhesion, and synaptic function [104]. Several investigations have demonstrated the participation of FYN in the development of seizures [105,106]. Individual genetic ablation of *Fyn* or *Tau* appears to be protective against aberrant excitatory neuronal activities in Alzheimer’s disease and epilepsy models [107], although it remains unclear whether the ablation of both *Fyn* and *Tau* would elicit more profound antiseizure and neuroprotective effects. However, in both cases and despite the proteomic data, we did not detect significant differences in the expression of *Fyn* and *Itch* mRNAs by RT-qPCR. In this case, the decrease was not low enough to generate a loss of critical function of the protein.

Thirdly, we studied the expression of the *Ralb* gene, selected for its relationship with the CXCR4 receptor pathway, and detected significantly at lower transcription levels in the GASH.sAUK.NR group. This data should be consistent with an increase in susceptibility to seizures (drop in the seizure threshold in accordance with that described in mice) [108]. This discrepancy leads us to think about the possible phenotypic differences between other rodent models with audiogenic epilepsy and the hamsters in our study.

*Ralb* encodes the Ras-like proto-oncogene B protein, a small GTPase that facilitates B cell migration by interacting with SDF-1 [109]. RALB is involved in the endocytic transport of tight junction components [110] and in the regulation of vesicular secretion dependent on different intracellular signals such as calcium concentration. This suggests that a decrease in Ralb expression could negatively affect the concentration of the so-called fast secretory reserve vesicles, which are located closest to the presynaptic membrane and are the first to respond following the entry of intracellular calcium [111]. Therefore, lower concentrations of these vesicles would decrease the release of neurotransmitters into the synaptic cleft. In addition, Bonham et al. [109], using bioinformatics tools, described a strong correlation of expression of *CXCR4* and *RALB*, among others. They detailed an altered expression of *CXCR4* and some of its correlated genes in the brain of patients with different neurodegenerative diseases such as frontotemporal dementia, Parkinson’s disease, or progressive supranuclear palsy, suggesting an important role of microglia and the immune system in neurodegenerative diseases [109].

Finally, other significant underexpressed pathways in the non-responder group were related to mitochondrial energy metabolism, the Krebs cycle, and ATP production. This suggests that reductions in mitochondrial activity and metabolic processes play a key role in protecting these animals from seizures. Specifically, the decreased expression of subunits of the electron transport chain (ETC), such as NDUFA2 and NDUFA13, points to lower Complex I activity, leading to diminished reactive oxygen species (ROS) production. A decrease in ROS could help preserve mitochondrial and neuronal function, preventing cellular stress that typically drives neuroexcitability and seizure initiation [112]. In addition, the pyruvate dehydrogenase complex, marked by reduced expression of PDHX, PDPR, and DLD, appeared less active in the non-responder group, reflecting a lower reliance on the glycolytic pathway and citric acid cycle. This alteration led to reduced lactate production and metabolic acidosis, conditions associated with neuronal hyperexcitability. The shift in pyruvate metabolism likely promotes an energy-efficient state, where alternative substrates such as fatty acids or ketone bodies are used, thereby preventing the hypermetabolic state seen in naïve animals [113]. Furthermore, the expression of proteins involved in fatty acid oxidation, including CPT2 and ACADSB, was reduced, indicating a decreased reliance on lipid metabolism. This leads to lower ATP production from fatty acids, but it also prevents mitochondrial overload and ROS generation, contributing to mitochondrial stability and protection from metabolic stress [114]. Additionally, proteins such as SLC25A1 and SLC25A12, involved in mitochondrial metabolite transport, were downregulated in the non-responder group. This reduction points to a lower metabolic rate, which reduces the excessive energy demands that could lead to neuronal hyperexcitability. By maintaining a more regulated mitochondrial function, these animals avoid the metabolic dysregulation seen in naïve animals, which contributes to seizure susceptibility, thus offering resistance to seizures [115].

#### 3.2.2. Weighted Correlation Network Analysis

Finally, we performed a weighted correlation network analysis and selected five candidates for further molecular validation: *Slc1a2*, *Slc1a3*, *Akt1*, *Cyfip2*, and *S100b*. *S100b* was the only transcript validated by molecular techniques, as we detected lower expression levels in the inferior colliculi of the sAUK.NR hamsters. S100B is a calcium-binding protein expressed mainly by astrocytes. It has been widely studied because of its prognostic and therapeutic potential, as it is considered a brain damage biomarker [116]. A dual role has been described for S100B depending on its concentration, where reduced levels of the protein promote neuronal survival and stimulate neurite outgrowth, while at elevated concentrations, S100B causes neuronal death [117]. Furthermore, different investigations have linked elevated serum [118] and brain [119] concentrations of S100B with the most severe cases of epilepsy. In our study, lower gene and protein levels of S100B were detected in the inferior colliculus of sAUK.NR hamsters compared to the naïve GASH/Sal group. Thus, this means that hamsters that suffer milder seizures diminish the expression of the S100B, probably to avoid its neurotoxic effects in the brain. We should analyze the gene’s expression pattern during the entire protocol to elucidate its possible function during seizure induction.

For the rest of the candidates, we did not detect significant changes for the *Slc1a3* and *Cyfip2* genes. In contrast to the proteomic results, *Slc1a2* and *Akt1* transcripts were at higher expression levels in the sAUK.NR group compared to the naïve one. This may be due to several reasons, such as that messenger and protein expression do not necessarily have to be directly correlated, the half-life of the protein, transcription factors expression, accelerated metabolism, etc.

## 4. Materials and Methods

### 4.1. Experimental Groups

For the purpose of this study, 41 GASH/Sal males were used. They were provided by the Experimental Animal Service of University of Salamanca. They were between 2 and 4 months old and were maintained in a controlled environment (21 ± 2 °C), with 12 h cycle of day and night, and with access ad libitum to food and water. All experiments complied with the guidelines for the use and care of experimental animals (RD 53/2013, Order ECC/566/2015, 20 March) and were approved by the Bioethical Committee of Salamanca University (approval No. 375).

### 4.2. Experimental Design

The GASH/Sal hamsters were divided into two groups: naïve and subjected to a sAUK process. In both groups, we performed analyses on the state of the auditory system or ABRs (before and after the stimulation protocol), studies on the inflammatory state, and proteomic studies of the epileptogenic nucleus. Euthanasia and sample collection were performed sixty minutes after the last exposure to sound. Based on the results of the proteomic analyses, we carried out bioinformatics and statistical procedures conducted for the selection of genes subsequently analyzed by RT-qPCR. A schematic diagram of the experimental design is shown in Figure 16.

Two sets of experiments were done to achieve a significant number in the results.

### 4.3. Audiogenic Kindling Procedure

In total, 21 GASH/Sal male hamsters were maintained at naïve conditions, without sound stimulation, and therefore, they did not suffer any seizures. Another subgroup of GASH/Sal (*n* = 20) hamsters were submitted to the audiogenic kindling process. It consisted of playing the sound stimuli three times a day for fifteen days at 9:00 a.m., 14:00, and 19:00 For that, animals were placed in a methacrylate cylinder and allowed to acclimate for one minute. Then, they were submitted to white noise of 120 dB SPL of intensity and 0–18 kHz of frequency. The stimulus was played until induction of the tonic–clonic phase or until one minute if the animals do not reach the tonic–clonic phase. Finally, animals remained for another minute post-stimulus in the cylinder. The sound was created using a high-pass filter (>500 Hz, Bruel & Kjaer (Nærum, Denmark) #4134 microphone and preamplifier #2619), digitized at 44.1 kHz and played by a computer-coupled amplifier (Fonestar MA-25T, Revilla de Camargo, Spain) and speaker (Beyma T2010, Valencia, Spain) located above the arena. Every stimulation was recorded for its visualization and analysis. For the classification of the seizure intensity, we followed the index described by Garcia-Cairasco et al. [18] (Table 2).

### 4.4. Study of the State of the Auditory System of Animals: ABRs

The status of the auditory system was measured in epileptic-prone animals at two timepoints, before starting the protocol, without experienced any convulsion, and twelve days from the beginning of the experiment at a time point at which the anesthesia used for the ABR would not interfere with the stimulation protocol.

We followed the protocol previously used in our laboratory [34]. In total, 8 GASH/Sal hamsters were anesthetized by 3% of isoflurane inhalation and maintained at 2%. Subsequently, three subcutaneous needle electrodes were placed at the vertex (reference electrode), the mastoid ipsilateral to the stimulated ear (active electrode), and the mastoid contralateral to the stimulated ear (ground electrode). Then, animals were placed in a close soundproofed field. For each ear, the procedure consisted in a 5-millisecond (ms) window, with a 1 ms prestimulus period, and a 0.1 ms alternating polarity click, with a repetition rate of 21 bursts/s, delivered in 10 dB ascending steps from 10 to 90 dB. Sounds were emitted by a magnetic speaker (TDT, Tucker-Davis Technologies, Multi-Field Magnetic Speakers, System RZ-6, Alachua, FL, USA) connected to the external auditory meatus of a single ear through a 10 cm long plastic tube. An ABR was obtained by averaging 1000 EEG responses to 1000 click stimuli. Evoked potentials were amplified and digitized using a Medusa RA16PA preamplifier and RA4LI head-stage. The final signal was filtered with a 500 Hz high-pass filter and a 3000 Hz low-pass filter. The ABR analysis was done using a custom-made script developed in MATLAB software (version R2014a, The MathWorks, Inc., Natick, MA, USA). The quality of each recording was assessed measuring the mean background voltage of the 1 ms period before the stimulus onset. The ABR threshold was defined as the stimulus level that evoked a mean voltage value greater than 2 times the standard deviation above the mean background activity. These auditory thresholds were confirmed by blind visual inspections at the lowest intensity, at which waves I and II were detectable above noise within the 5 ms response window immediately before the stimulus onset. The amplitude and latency of click ABR waves were measured at the suprathreshold hearing level of 90 dB SPL, analyzing positive and negative peaks of each wave with the MATLAB program. Wave amplitude was defined as the sum of the positive and negative peak values of each wave. After the experiment, animals were placed at a heating pad to maintain a constant body temperature of 37 °C for its recovery.

### 4.5. Blood Processing and Cytokine Arrays

Blood samples of GASH.naïve (*n* = 6) and GASH.sAUK.NR hamsters (*n* = 4) were obtained from the subclavian vena and quickly processed. First, they were centrifuged for ten minutes at 4000 rpm, and the resultant supernatant was again centrifuged for twenty minutes at 3000 rpm, both at room temperature. Plasma was stored at −20 °C until use. The Mouse Neuro Antibody Array (Abcam (Cambridge, UK), cat. No. ab211069) was used for cytokine plasma levels detection. As indicated for the manufacture, it detects up to twenty-three cytokines: GCSF (Granulocyte Colony-Stimulating Factor), IGF-1 (Insulin-like Growth Factor 1), IFN-gamma (Interferon-gamma), IL-1 alpha (Interleukin-1 alpha), IL-1 beta (Interleukin-1 beta), IL-4 (Interleukin-4), IL-6 (Interleukin-6), IL-10 (Interleukin-10), KC (Keratinocyte Chemoattractant), LIX (LPS-induced CXC Chemokine), MCP-1 (Monocyte Chemoattractant Protein 1), M-CSF (Macrophage Colony-stimulating Factor), MIP-1 alpha (Macrophage Inflammatory Protein 1 alpha), RAGE (Receptor for Advanced Glycation End-products), SDF-1 (Stromal Cell-Derived Factor 1), TARC (Thymus and Activation-Regulated Chemokine), TNF-alpha (Tumor Necrosis Factor alpha) and VEGF-A (Vascular Endothelial Growth Factor A). Two proteins, Matrix Metalloproteinases 2 and 3 (MMP2 and MMP3), were excluded from the study due to the use of EDTA as an anticoagulant. First, arrays were blocked for 30 min at room temperature, followed by an overnight incubation with undiluted plasma at 4 °C. Then, membranes were washed with Wash Buffer I, Wash Buffer II, and finally incubated with the Biotinylated Antibody Cocktail at 4 °C overnight. Once incubation time elapsed, membranes were washed with Wash Buffer I and Wash Buffer II as the previous day. Subsequently, membranes were incubated overnight at 4 °C with 1X HRP-Streptavidin solution. Last, HRP-Streptavidin solution was eliminated and membranes washed. Finally, membranes were incubated with a mixture of detection buffers C and D (1:1) to visualize the chemiluminescence signals for five seconds. Averaged intensity of each spot was quantified with ImageJ 2.9.0 (NIH, Bethesda, MD, USA) [120], and normalization was related to each membrane’s positive control.

### 4.6. Euthanasia and Tissue Recollection for Molecular Studies

To obtain the epileptogenic foci, one hour after the last stimulation, animals were deeply anesthetized, and once areflexia was verified, they were decapitated. Samples were snap-frozen in liquid nitrogen and stored at −80 °C until use. For expression change experiments, RNA was extracted following the TRIzol^TM^ Reagent protocol (Thermo Fisher Scientific (Waltham, MA, USA), cat No. 15596026) for RNA isolation. Samples were homogenized in 1 mL of TRIzol^TM^ per 50–100 mg of tissue and then incubated for 5 min at room temperature. Next, 0.2 mL of chloroform were added for each mL of TRIzol^TM^ used. The mixture was incubated for 2–3 min and then centrifuged for 15 min 12,000× *g* at 4 °C. For the RNA precipitation, 0.5 mL of isopropanol was added to the aqueous phase for each milliliter of TRIzol^TM^ previously used for lysis. The preparation was then centrifuged for 10 min 12,000× *g* at 4 °C. The pellet obtained was resuspended in 1 mL of 75% ethanol per mL of TRIzol^TM^ and centrifuged for 5 min 7500× *g* at 4 °C. After discarding the supernatant, the final pellet was dried under the laminar flow hood. Finally, it was resuspended in 20 μL of RNAse-free water and incubated for 10 min in a water bath at 56 °C. Concentration and purity (260/280 ratio) of isolated RNA were measured by the Nanodrop^TM^ 2000c. Total RNA was submitted to a revert transcription reaction following the Thermo Scientific^TM^ RevertAid^TM^ First Strand cDNA Synthesis Kit. According to the protocol, up to 5 μg of the RNA can be mixed with 1 μL of primer Oligo (dT)18 for a final volume of 12 μL. The mixture ran a cycle of 5 min at 65 °C in the cycler c1000^TM^ Thermal Cycler (BIO-RAD, Hercules, CA, USA). Next, a mix was made that consisted of 4 μL of reaction buffer, 1 μL of the RiboLock RNase Inhibitor (20 U/μL), 2 μL of dNTPs 10 mM, and 1 μL of the transcriptase enzyme RevertAid M-MuLV RT (200 U/μL), and the new mixture went through a cycle of 60 min at 42 °C and 5 min at 70 °C. The obtained cDNA was then subjected to RT-qPCR.

### 4.7. Mass Spectrometry Sample Preparation

IC samples were processed for protein extraction. DDA (data-dependent acquisition) and DIA methods were followed as described in Garcia-Peral et al. [121]. Briefly, proteins were obtained following several steps of sample lysis, reduction of disulfide bond by dithiothreitol (DTT), and protein precipitation using chilled acetone with 10% TCA (*v*/*v*). The pellet obtained from these steps was lysed again, sonicated, and centrifuged to obtain the final supernatant with the proteins. Such proteins were quantified with the Bradford method. To obtain the peptides, equal amounts of proteins were hydrolyzed with trypsin, and afterward, peptides were desalted applying a Strata X SPE desalting column (Strata X 33 μm polymeric reversed-phase column; Phenomenex, Torrance, CA, USA). Again, DTT and iodoacetamide (IAM) were used to reduce disulfide bonds and alkylate cysteines, respectively.

Peptides from each sample were mixed, diluted, and separated by its hydrophobicity following several column-based affinities and elution rates to obtain a total of 10 fractions. The ten high pH fractions were subjected to DDA and the unfractionated to DIA.

As indicated in previous work [121], BGI company (Shenzhen, China) oversaw protein identification analysis. The same DIA protocol, DDA spectral library, and databases were used in this work.

### 4.8. Proteomic Data Processing

This procedure relied on the sample data produced by high-resolution mass spectrometry (MS). Peptides and proteins were quantified with the MSstats 4.14.1. software package [122], which accounted for intra-system error corrections and normalization across all samples.

The databases containing the log abundance values for each protein in each sample were analyzed using RStudio software 4.4.2 for each batch. The pipeline involves first filtering the log abundance values for missing data (NA). The filter criteria require that each protein be present within each group at least 60% of the time. Additionally, 3 NA values were allowed for each protein in the first batch and 2 in the second batch. After cleaning the NAs, the next step was to impute the sample mean for the remaining NAs in the dataset. Both datasets were subset by each replicate, and if there was a missing value, it was replaced with the sample mean.

Proteins present in both experiments were considered for further investigation. Given the high sensitivity and variability of the Q-Exactive HF X equipment in performing shotgun proteomics, batch effects were highly likely. We utilized a batch effect correction algorithm using the sva RStudio package 3.52.0 [123] based on the parametric empirical Bayes framework. The correction applies only to the mean of the batch effect without scaling.

### 4.9. Proteomic Data Analyses for Uncovering Potential Status Epilepticus Regulation in GASH/Sal Model

#### 4.9.1. Unsupervised Analysis

The Uniform Manifold Approximation and Projection (UMAP) unsupervised analysis was performed using the entire and a reduced dataset that was conformed with the DEPs obtained in the comparison GASH.naïve vs. GASH.sAUK.NR. Briefly, the Seurat RStudio package 5.1.0 [124] was used to normalize the data using a relative-to-maximum approach, followed by scaling and centering the data, identifying clusters through shared nearest neighbor (SNN) modularity optimization, and reducing dimensions with the Uniform Manifold Approximation and Projection (UMAP) algorithm.

#### 4.9.2. Univariate *t*-Test Analysis

Univariate statistical analyses were applied to seek differential expressed proteins (DEPs) between the GASH.sAUK.NR and GASH.naive groups. A *t*-test was mapped robustly within each protein and between groups using the purrr R package 1.0.2 [125], and the cut-off value was set at 0.05. Also, the dataset was group split; within both groups, the mean and standard deviation (SD) of the abundance values were calculated for each protein. Subsequently, the mean abundance differences (Fold Change) were computed for each protein with the pooled standard deviation, the square root of the sum of square SD (SD GASH.naïve and SD GASH.sAUK.NR) divided by 2. The Cohen’s effect size, ref. [126] the Fold Change divided by the pooled standard deviation, was computed as a statistical effect size measurement. The cut-off value was set at 2 for upregulated proteins and −2 for downregulated proteins.

Additionally, UniProt identifiers from *Mesocricetus auratus* were converted to *Homo sapiens* gene symbols using the Uniprot database [127] and the RStudio package gprofiler2 (version 0.2.3) [128]. Proteins with no isoforms were discarded. Finally, some isoforms were detected in our predefined set of proteins; for this reason, only the most statistically significant protein isoform remained for downstream analyses.

#### 4.9.3. GSEA, Data Visualization, and Interpretation

GSEA was performed to determine whether our defined set of proteins showed statistically significant differences between the two biological states (GASH.sAUK.NR vs. GASH.naïve).

Extraction of value information from GSEA comprises exploring curated biological databases. Many of them are available for human and mouse species; however, the lack of these for other laboratory species remains a problem. Consequently, *Mesocricetus auratus* human orthologous genes symbols were used as input.

GSEA was conducted following the method described in Reimand et al. Version 4.3.3 of GSEA from Broad Institute [129,130] was used to obtain enriched pathways in ranked gene list mode. The rank of each gene was calculated by multiplying −log10 *p*-value by the sign of CES. Subsequently, the list was launched against the database of all curated gene sets from the human collection of the Molecular Signature Database (MSigDB) [129,131] with the following parameters. The number of permutations was set to 1000 (it specifies the number of times that the gene sets will be randomized to calculate the FDR value), and the maximum size of pathways was set to <200 to exclude larger sets, as they are overly general, and they do not contribute to the interpretability of results. Additionally, the minimum size of pathways was set to 15, as it is redundant with larger pathways.

Finally, GSEA generated all positive and negative regulated pathways between biological groups. A pathway was considered differentially expressed if the FDR value was equal to or below the significance threshold (0.05).

Furthermore, pathway visualization plays an important role in simplifying inherently redundant information. Databases are organized hierarchically, as they include specific and general pathways with many shared genes. Consequently, collapsing redundant pathways into a single biological theme simplifies interpretation [132].

Statistically significant pathways from GSEA were visualized with EnrichmentMap (version 3.4.0) [19], an application of Cytoscape (version 3.10.2) [133], which allows for gene sets (pathways and ontology terms) organization into networks. In this way, mutually overlapping gene sets are clustered together. Additionally, PathVisio (version 3.3.0) [20] was used to visualize curated epilepsy-related pathways available in WikiPathways [134]. This software enables the user to analyze (coloring, editing, and drawing) biological pathways according to provided omics data (e.g., protein expression). Finally, top 3 genes with highest ES belonging to significant epilepsy-related pathways were screened for validation.

#### 4.9.4. Network Analysis

The WGCNA algorithm, available in the R package named WGCNA (version 1.73) [25,26], was adopted to find modules potentially correlated with epilepsy status. Modules are clusters of proteins with highly correlated expression patterns across the samples. A weighted adjacency matrix was first calculated using pairwise correlations between expression data and soft thresholding. To ensure a scale-free network, the pickSoftThreshold function was implemented to choose an optimal soft thresholding power ranging from 1 to 50. The selection of the power was based on scale independence and mean connectivity measures. The chosen power value was set to 30 (Figure 14A,B). Subsequently, the Topological Overlap Matrix (TOM) was built based on the mentioned adjacent matrix. Then, the function blockwiseModules was employed for automatic module detection. Briefly, this function first clusters genes hierarchically based on their pairwise similarity. This produces a dendrogram that reflects the clustering structure of the genes. Afterwards, the Dynamic Tree Cut method was used to cut the resulting dendrogram and define initial gene modules. The final merging step fuse modules which are highly correlated based on their module eigenprotein similarity (Figure 14C). The module eigenprotein represents the maximum variance within a module, which is a linear combination of the protein expression profiles. Additionally, using eigenprotein delivers a summary statistic for each module and certainly compensates for the need for stringent multiple testing corrections.

Next, we correlate eigenproteins with the trait (epilepsy status) to identify significant module–epilepsy associations. Moreover, we quantified protein significance, which establishes the correlation between proteins and the trait and module membership, which determines the correlation of eigenproteins and the protein expression profile.

Eventually, significant proteins (*p*-value < 0.05) of significant modules (*p*-value < 0.05) were submitted to the String database [27] to seek the protein–protein interaction network. An interaction high confidence score cut-off value of 0.7 was chosen, and the k-means clustering algorithm was implemented for subsetting the larger subnetwork (higher number of nodes).

Finally, proteins belonging to the subnetwork were analyzed in Disease 2.0 [28] and Disgenet [29] databases for checking genes for epilepsy known relevance. Disgenet includes database sources classified as Curated, Animal models, Inferred, and Literature, while Disease 2.0 extracts information from curated databases, genome-wide association studies (GWAS), and automatic text mining of the biomedical literature. GDA (Gene–Disease Association) score from Disgenet and Confidence Score from Disease 2.0 were extracted (the highest score term linked to epilepsy was selected). Then, the Disease 2.0 score was normalized, and the mean between database scores was calculated to rank every protein. Top 5 proteins with the highest scores were screened for RT-qPCR validation.

### 4.10. Real-Time Quantitative Reverse Transcription PCR

The cDNA obtained (explained in Section 4.6 *Euthanasia and Tissue Recollection for Molecular Studies*) was then subjected to several RT-qPCRs using the Quant Studio 7^TM^ Flex Real-Time PCR System (Applied Biosystem by Life Technologies, Carlsbad, CA, USA). Each reaction contained a mix of 7 μL of SYBER Green, 0.8 μL of each pair of primers (Table 3), 10 ng of cDNA, and water up to a final volume of 20 μL. The cycle conditions consisted of 10 min at 95 °C, followed by 40 cycles of 15 s at 95 °C, 30 s at 60 °C, and a final elongation of 3 min at 72 °C. For each gene, between 6 and 9 animal samples from both experimental groups were used. Moreover, for each sample, gene expression level was measured for triplicates, and the β-actin was used as a housekeeping gene. To analyze the results, the 2^−ΔΔCT^ method was used [135].

## 5. Conclusions

All the results presented in this work indicate that there is a subpopulation of hamsters that responds to convulsive stress by generating mechanisms that compensate for the damage caused by the seizures. The plasma elevation of proinflammatory cytokines (IL-1; IL-6; TNF-alpha; INF-gamma), which is traditionally associated with epileptic seizures, is countered by the elevation of anti-inflammatory mechanisms (IL-10; Fractalkine; GCSF) and the decrease in plasma concentration of IGF-1. Consistently, the upregulated proteins present a neuroprotective profile (Cpe, Cpne5, Vgf, G3bp2, Gpc1, Ikbkg, and Sdc3) or are also linked to the stimulation of neuronal survival (Agap2, Dpp8, and Madd). In the same line of evidence is the decrease in S100b, suggesting that the rise in threshold of seizures induced by sound stimuli in the GASH.sAUK.NR group is due to a general compensatory phenomenon that prevents the spread of damage to the limbic structures.

## Figures and Tables

**Figure 1 ijms-26-02331-f001:**
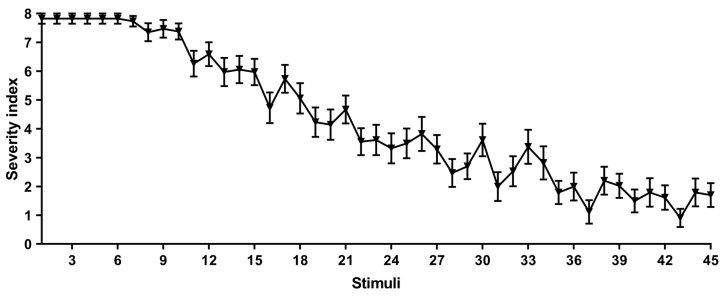
Progression of audiogenic seizure severity of GASH/Sal hamsters submitted to the sAUK protocol.

**Figure 2 ijms-26-02331-f002:**
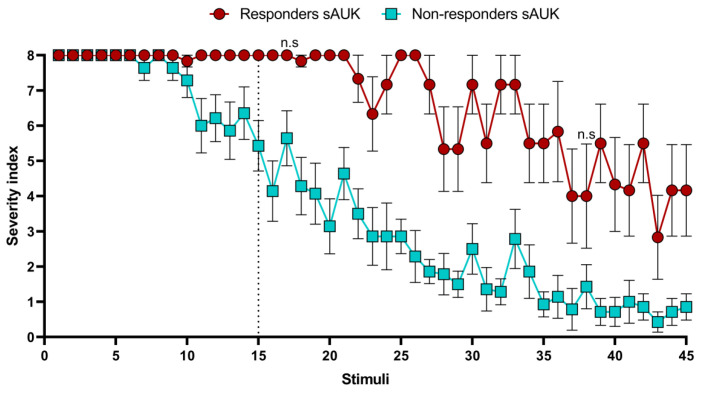
Severity index progression of the two groups, responder and non-responder hamsters, during the fifteen-day protocol. From the 15th stimulation, differences in the mean severity index between the two groups were detected (*p* < 0.05) except for the 17th and 38th stimulus, at which no differences were detected (non-significant, n.s). Each point represents the mean ± SEM (error bars) of the severity index.

**Figure 3 ijms-26-02331-f003:**
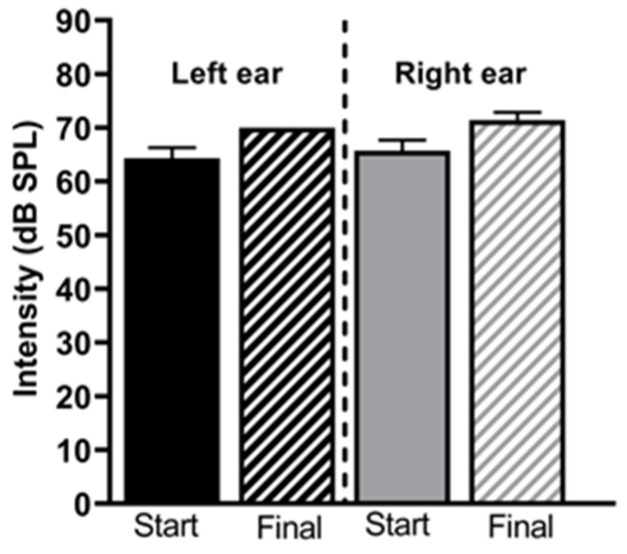
Threshold analysis for stimulated hamsters. Before the start and the last day of the protocol, GASH/Sal hamsters were submitted to ABRs. No significant differences were detected for the mean threshold between t0 and final time, neither for left (*p* = 0.07) nor right (*p* = 0.122) ear with the Mann–Whitney test (*n* = 7). Graphs display the mean with SEM error bars.

**Figure 4 ijms-26-02331-f004:**
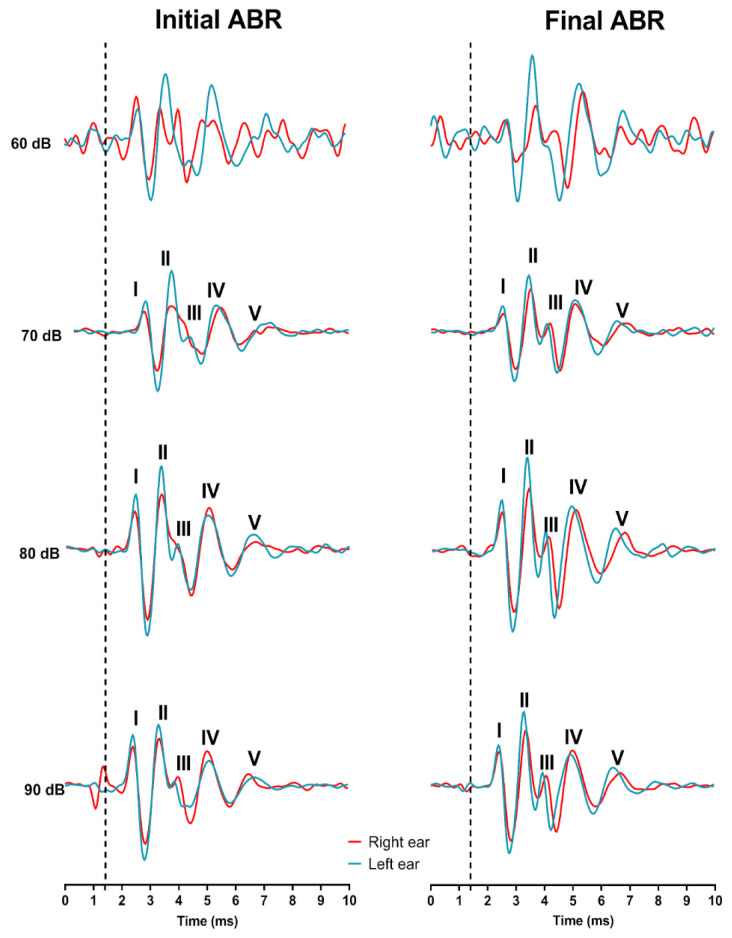
Representation of the average of the ABR waves for ascending intensities (60 to 90 dB). ABR waveforms obtained after click stimulation from the animals analyzed before (**left side**) and after the audiogenic kindling (**right side**). Plot shows ABR amplitudes in microvolts (mV) for each waveform response (I, II, III, IV, and V) measured at 90 dB SPL. Wave I corresponds to the response of the auditory nerve, wave II to the cochlear nuclei, wave III is associated with the superior olivary complex, wave IV corresponds to the lateral lemniscus and inferior colliculus, and finally, wave V refers to the response of the medial geniculate body. It is important to note that no significant differences were observed between the initial (t0) and final time points in the left ear (blue) nor in the right ear (red) of the audiogenic kindling.

**Figure 5 ijms-26-02331-f005:**
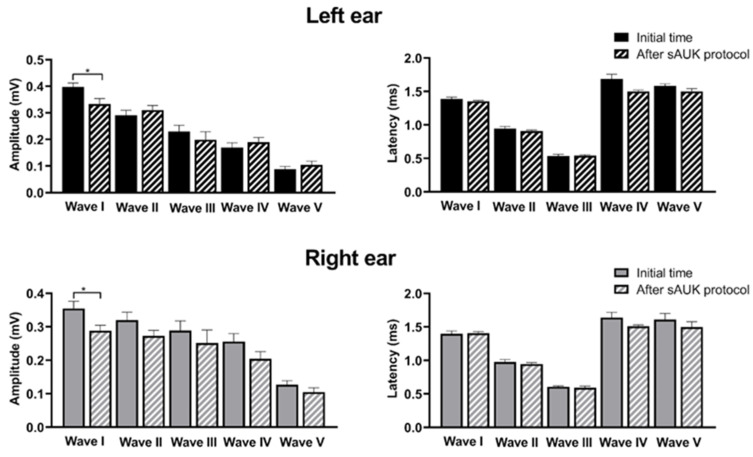
Amplitude and latency analysis before and after the sAUK protocol. For both ears at the end of the protocol, the amplitude of wave I was significantly lower than before the protocol (*p* < 0.05), without significant changes in the amplitude of the rest of the waves. No significant changes were detected for any latencies measured for left and right ears. Each bar in the histograms represents mean ± SEM. * *p*-value < 0.05.

**Figure 6 ijms-26-02331-f006:**
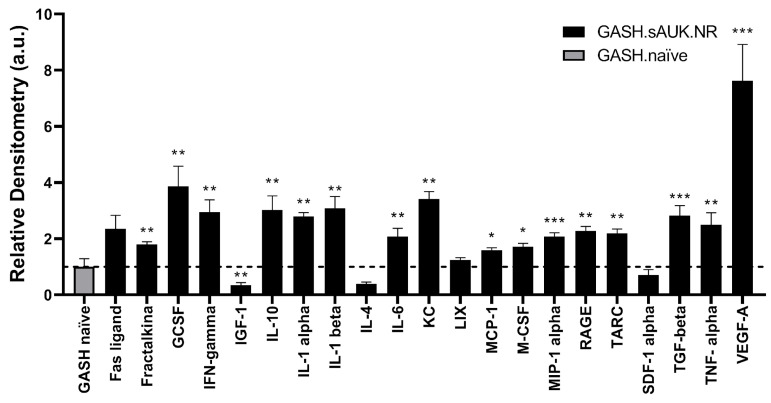
Differences in cytokine levels in blood between GASH/Sal naïve hamsters (*n* = 6) and stimulated non-responder hamsters (*n* = 4). We detected higher protein levels in almost all the cytokines included in the array except for the Insulin-like Growth Factor 1 (IGF-1). * *p*-value < 0.05; ** < 0.01; *** < 0.001. Dashed line indicates the normalization level related to the different cytokines in the GASH naïve hamsters. Depending on the normality of the samples, unpaired *t*-test or Mann–Whitney test was used. Each bar in the histograms represents mean ± SEM. Abbreviations: GCSF (Granulocyte Colony-Stimulating Factor), IFN-gamma (Interferon-gamma), IGF-1 (Insulin-like Growth Factor 1), IL-1 alpha (Interleukin-1 alpha), IL-1 beta (Interleukin-1 beta), IL-4 (Interleukin-4), IL-6 (Interleukin-6), IL-10 (Interleukin-10), KC (Keratinocyte Chemoattractant), LIX (LPS-induced CXC Chemokine), MCP-1 (Monocyte Chemoattractant Protein 1), M-CSF (Macrophage Colony-Stimulating Factor), MIP-1 alpha (Macrophage Inflammatory Protein 1 alpha), RAGE (Receptor for Advanced Glycation End-products), SDF-1 (Stromal Cell-Derived Factor 1), TARC (Thymus and Activation-Regulated Chemokine), TGF-beta (Transforming Growth Factor beta), TNF-alpha (Tumor Necrosis Factor alpha), VEGF-A (Vascular Endothelial Growth Factor A).

**Figure 7 ijms-26-02331-f007:**
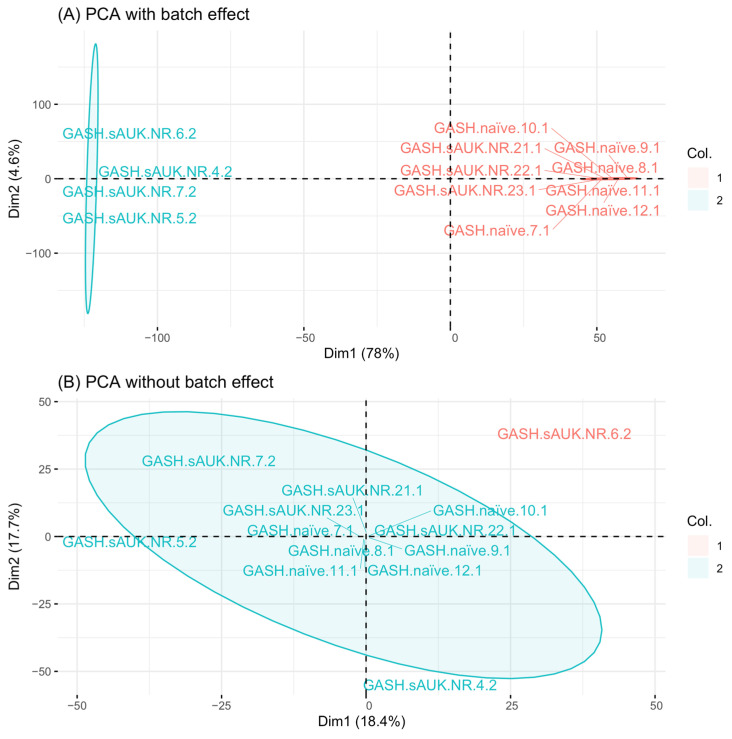
Principal component analysis considering all proteins with and without batch effect. The x axis and y axis represent the first and second components, respectively. (**A**) PCA with batch effect. Animals NR-4, NR-5, NR-6, and NR-7 (green) correspond to experiment 2, while all naïve animals plus NR-21, NR-22, and NR-23 (red) proceeded in experiment 1. (**B**) PCA without batch effect. A total of 91.6% (11/12) of the animals fall within the same cluster (green).

**Figure 8 ijms-26-02331-f008:**
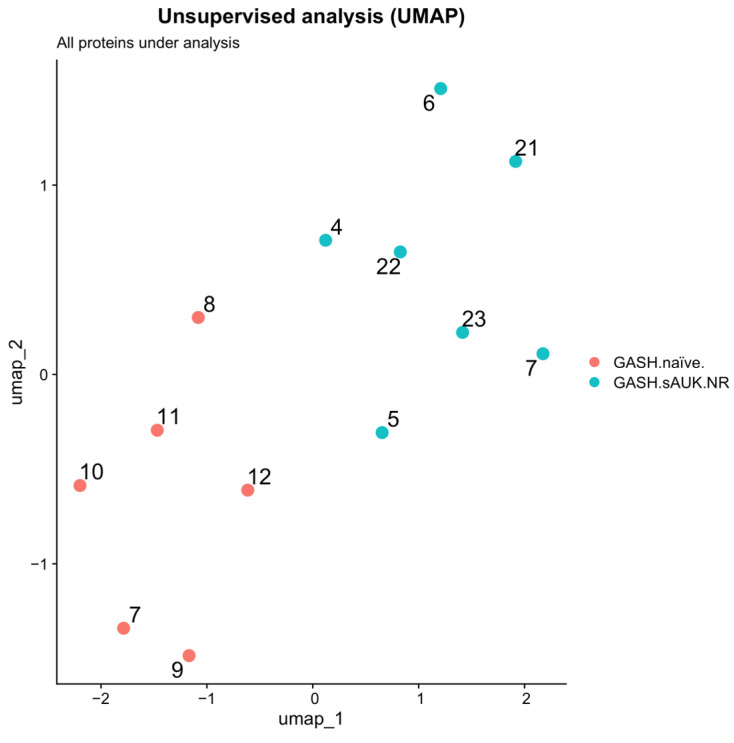
Uniform Manifold Approximation and Projection (UMAP) dimension plot. Both plots represent dimensions 2 and 1 of the UMAP algorithm on the y axis and the x axis, respectively. The plot was constructed with all detected proteins.

**Figure 9 ijms-26-02331-f009:**
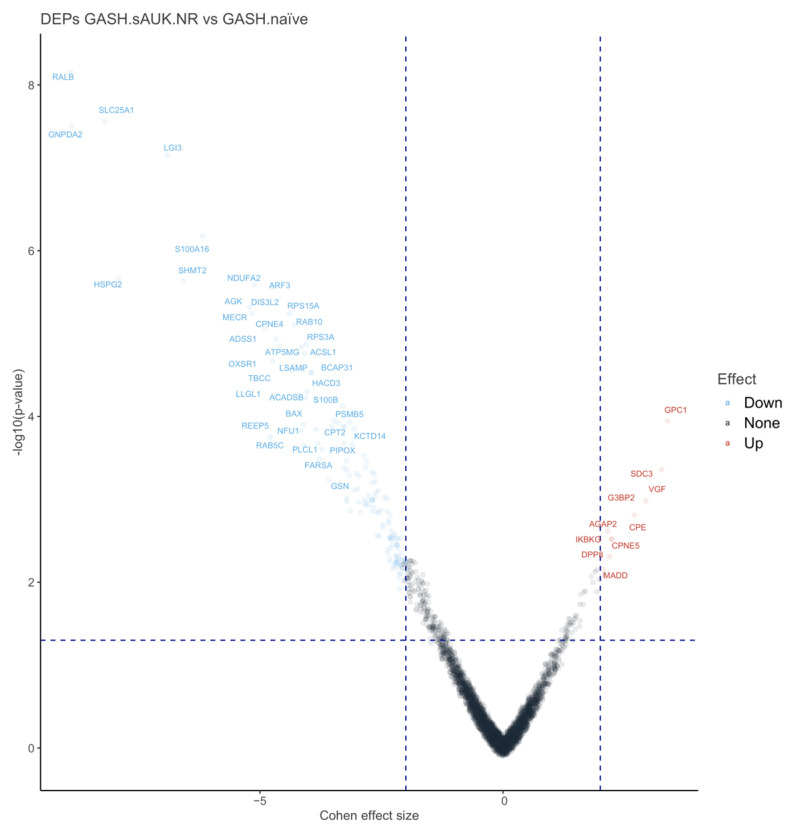
Volcano plot of DEPs GASH.sAUK.NR vs. GASH.naïve. The volcano plot represents the y-axis and x-axis −log10 (*p*-value) and the Cohen effect size, respectively.

**Figure 10 ijms-26-02331-f010:**
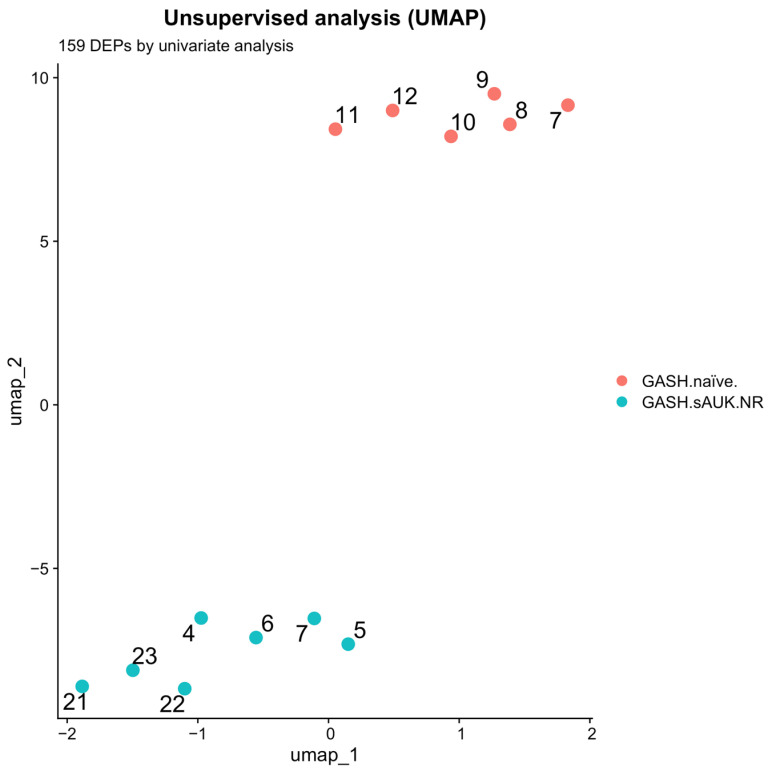
Uniform Manifold Approximation and Projection (UMAP) dimension plot. The plot represents dimensions 2 and 1 of the UMAP algorithm on the y axis and the x axis, respectively. The plot was constructed with all DEPs detected in the univariate analysis.

**Figure 11 ijms-26-02331-f011:**
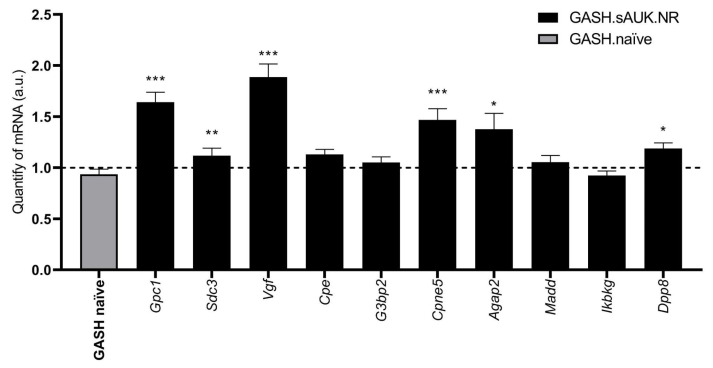
Gene expression differences of genes encoding the overexpressed DEPs in the inferior colliculus of GASH.sAUK.NR vs. GASH.naïve. Histogram shows relative quantities of transcripts of *Gpc1*, *Sdc3*, *Vgf*, *Cpe*, *G3bp2*, *Cpne5*, *Agap2*, *Madd*, *Ikbkg*, and *Dpp8*. The relative mRNA expression of each gene was normalized to *β-actin*. Each bar in the histograms represents mean ± SEM. Asterisks indicate significant differences between experimental groups (* *p* < 0.05, ** *p* < 0.01, *** *p* < 0.001). Dashed line indicates the normalization level related to the different transcripts in the GASH naïve hamsters. Abbreviations: *Agap2*: Arf-GAP with GTPase domain, ANK repeat, and PH domain-containing protein 2; *Sdc3*: Syndecan 3; *Cpe*: Carboxypeptidase E; *Cpne5*: Copine 5; *Dpp8*: Dipeptidyl Peptidase 8; GASH.sAUK.NR: non-responder GASH/Sal submitted to audiogenic kindling; *G3bp2*: GTPAse Activating Protein (SH3 Domain) Binding Protein 2; *Gpc1*: Glypican-1; *Ikbkg*: Inhibitor of kappaB Kinase Gamma; *Madd*: MAP Kinase Activating Death Domain; *Vgf*: Nerve Growth Factor Inducible.

**Figure 12 ijms-26-02331-f012:**
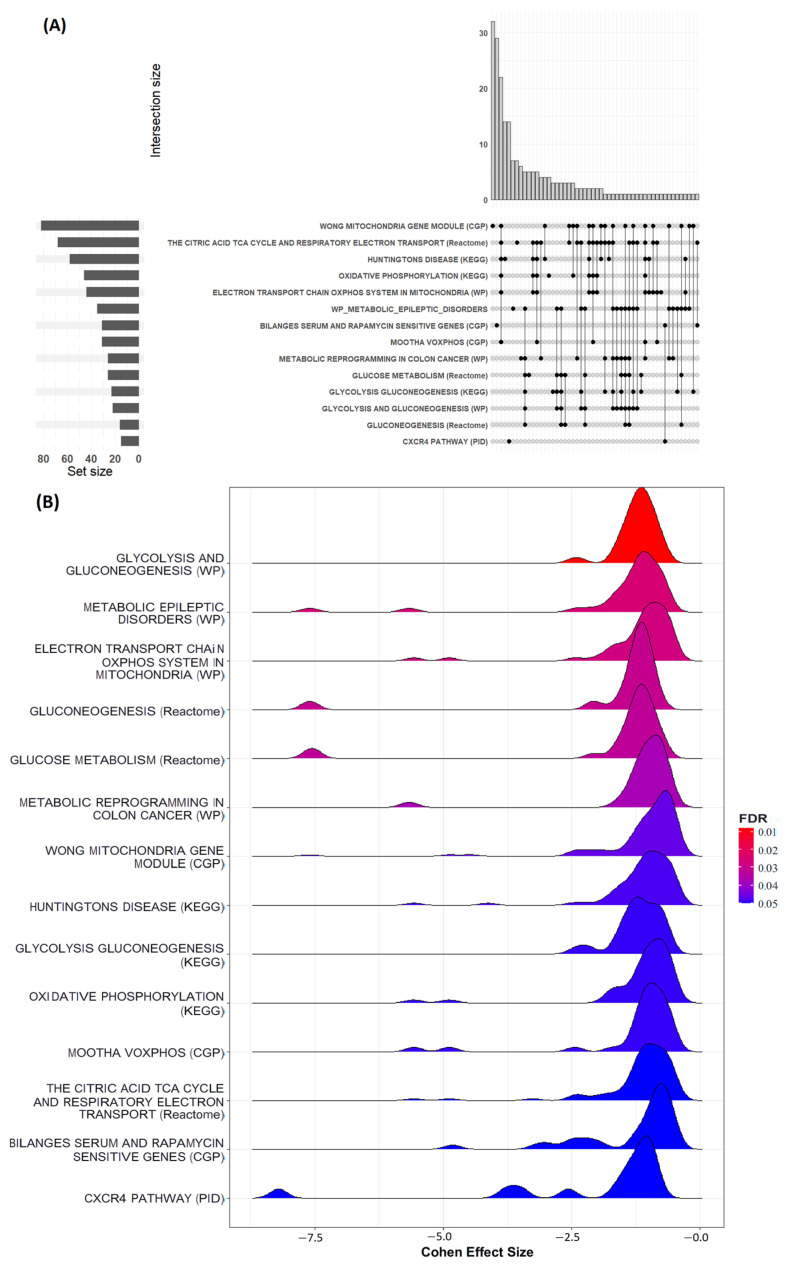
(**A**) UpSet plot. Enriched terms showed high semantic similarity and many shared genes except for “Bilanges serum and rapamycin sensitive genes” and “CXCR4 pathway”. The plot was constructed with the R package ComplexUpset (version 1.3.5) [22,23]. (**B**) Ridge plot. The density distributions of all enriched terms exhibited peak frequency values clustered around −1.75 CES. The top enriched downregulated pathway in GSEA was “Glycolysis and gluconeogenesis”. The plot was constructed with the R package enrichplot (version 1.26.6) [24]. Abbreviations of source databases: CGP (Chemical and genetic perturbations from the Human Molecular Signatures Database), KEGG (Kyoto Encyclopedia of Genes and Genomes), PID (Pathway Interaction Database), and WP (Wikipathways).

**Figure 13 ijms-26-02331-f013:**
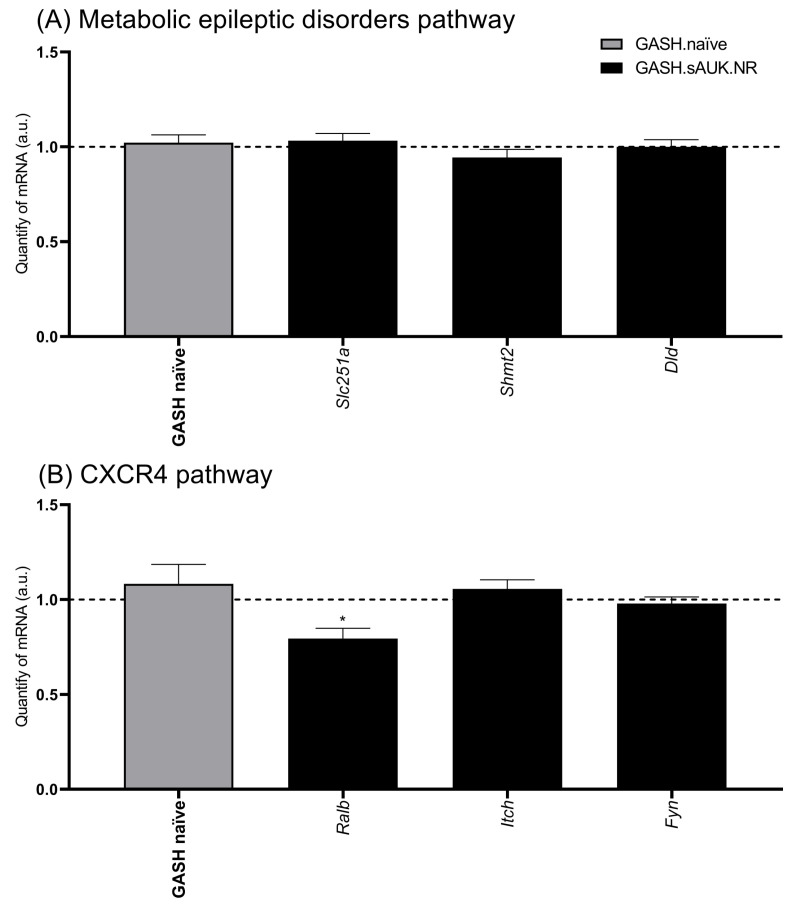
Inferior colliculus expression changes of GSEA-selected genes. (**A**) Histograms show relative quantities of transcripts from the metabolic epileptic disorders pathway: *Slc25a1*, *Shmt2*, *Dld* in the GASH.sAUK.NR compared to GASH.naïve hamsters. (**B**) Histograms show relative quantities of transcripts from the CXCR4 pathway: *Ralb*, *Itch*, and *Fyn* in the GASH.sAUK.NR compared to GASH.naïve hamsters. The relative mRNA expression of each gene was normalized to *β-actin*. Each bar in the histograms represents mean ± SEM. Asterisks indicate significant differences between experimental groups (* *p* < 0.05). Dashed line indicates the normalization level related to the different transcripts in the GASH naïve hamsters. Abbreviations: CXCR4: C-X-C: Chemokine Receptor Type 4; *Dld*: Dihydrolipoamide Dehydrogenase; *Fyn*: FYN Proto-Oncogene, Src Family Tyrosine Kinase; GASH.sAUK.NR: non-responder GASH/Sal submitted to audiogenic kindling; *Itch*: Itchy E3 Ubiquitin Protein Ligase; *Ralb*: RAS-Like proto-oncogene B; *Shmt2*: Serine Hydroxymethyltransferase 2; *Slc25a1*: Solute Carrier Family 25 Member 1.

**Figure 14 ijms-26-02331-f014:**
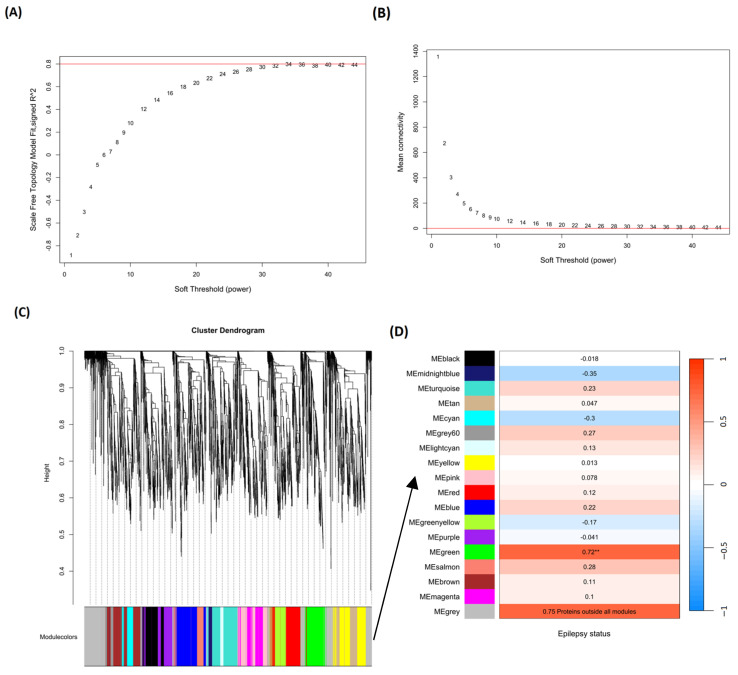
Scale independence (**A**) and mean connectivity (**B**) of different soft-thresholding values. The chosen value was set at 30. (**C**) Merged modules dendrogram. Clusters were constructed based on the TOM matrix using hierarchical clustering and the Dynamic Tree Cut method. Modules whose eigenprotein correlation was above 0.25 were merged. The minimum module set size was established at 30. (**D**) Module–epilepsy association. The heatmap illustrates the 17 module correlations with epilepsy status. Module gray is excluded considering that it includes genes with no module associations. Module green (MEgreen) was significant. ** *p*-value < 0.01.

**Figure 15 ijms-26-02331-f015:**
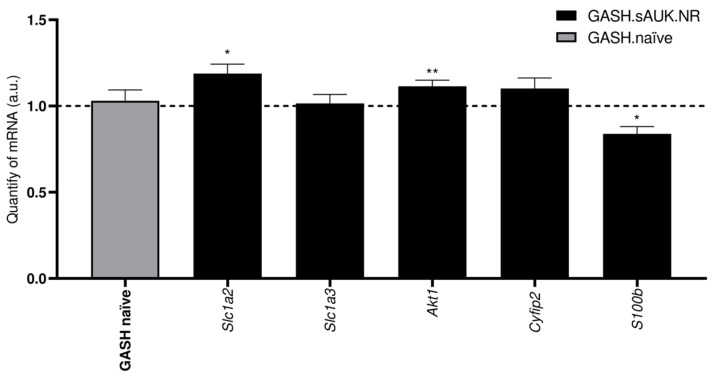
Gene expression changes of WGCNA-selected candidates in the inferior colliculus of GASH.sAUK.NR vs. GASH.naïve. Histograms show relative quantities of transcripts *Slc1a2*, *Slc1a3*, *Akt1*, *Cyfip2* and *S100b*. The relative mRNA expression of each gene was normalized to *β-actin*. Each bar in the histograms represents mean ± SEM. Asterisks indicate significant differences between experimental groups (* *p* < 0.05, ** *p* < 0.01). Dashed line indicates the normalization level related to the different transcripts in the GASH naïve hamsters. Abbreviations: *Akt1*: RAC-alpha serine/threonine-protein kinase; *Cyfip2*: Cytoplasmic FMR1 Interacting Protein 2; *S100b*: S100 calcium-binding protein B; GASH.sAUK.NR: non-responder GASH/Sal submitted to audiogenic kindling; *Slc1a2*: Solute Carrier Family 1 Member 2; *Slc1a3*: Solute Carrier Family 1 Member 3.

**Figure 16 ijms-26-02331-f016:**
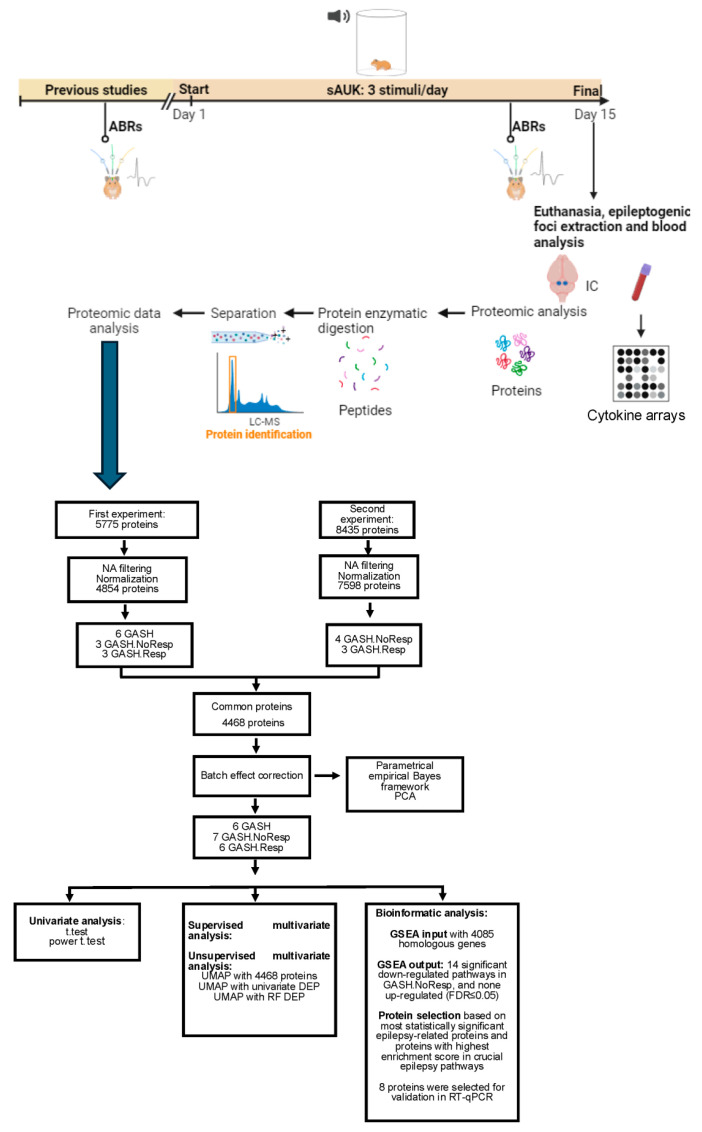
Experimental design. It includes the auditory brainstem response test performed before the stimulation protocol, the proper sAUK protocol, and the bioinformatic and statistics procedures conducted for the selection of genes subsequently analyzed by RT-qPCR.

**Table 1 ijms-26-02331-t001:** Top five proteins with the highest epilepsy scores.

Protein Symbol	Protein Name	Epilepsy Score
SLC1A2	Excitatory Amino Acid Transporter 2	0.800
CYFIP2	Cytoplasmic FMR1-interacting Protein 2	0.700
SLC1A3	Excitatory Amino Acid Transporter 1	0.693
AKT1	RAC-alpha serine/threonine-protein kinase	0.677
S100B	S100 calcium-binding protein B	0.527

**Table 2 ijms-26-02331-t002:** Categories of the seizure’s severity. Modified from Cairasco et al., 1996 [18].

Seizure Index		
SI	Seizure Behaviors	Score
0.00	No seizures	0
0.11	One wild running	1
0.23	One wild running (running plus jumping plus atonic fall)	2
0.38	Two wild runnings	3
0.61	Tonic convulsion (opisthotonus)	4
0.85	Tonic seizures plus generalized clonic convulsions	5
0.90	Head ventral flexion plus cSI5	6
0.95	Forelimb extension plus cSI6	7
1.00	Hindlimb extension plus cSI7	8
Categories, which are generally followed by hindlimb clonic convulsions (CCV2)		

**Table 3 ijms-26-02331-t003:** Pair of primers used for each gene for the amplifications in the PCR procedure. Abbreviations: *Agap2*: Arf-GAP with GTPase domain, ANK repeat, and PH domain-containing protein 2; *Akt1*: RAC-alpha serine/threonine-protein kinase 1; *β-act*: β-actin; *Cpe*: Carboxypeptidase E; *Cpne5*: Copine 5; *Cyfip2*: Cytoplasmic FMR1-interacting Protein 2; *Dld*: Dihydrolipoamide Dehydrogenase; *Dpp8*: Dipeptidyl Peptidase 8; *Fyn*: FYN Proto-Oncogene, Src Family Tyrosine Kinase; *Gpc1*: Glypican-1; *G3bp2*: GTPase Activating Protein (SH3 Domain) Binding Protein 2; *Ikbkg*: Inhibitor of kappaB Kinase Gamma; *Itch*: Itchy E3 Ubiquitin Protein Ligase; *Madd*: MAP Kinase Activating Death Domain; *Ralb*: RAS-like proto-oncogene B; *Sdc3*: Syndecan 3; *Shmt2*: Serine Hydroxymethyltransferase 2; *Slc1a2*: Excitatory Amino Acid Transporter 2; *Slc1a3*: Excitatory Amino Acid Transporter 1; *S100b*: S100 calcium-binding protein B; *Slc25a1*: Solute Carrier Family 25 Member 1; *Vgf*: Nerve Growth Factor Inducible.

Amplicon Size and Primers Used for the RT-qPCR Experiments		
Gene Amplified	Sequence (5′-3′)	Amplicon (pb)
*Gpc1*	GCTGCCGACTGATGACTACT	214
	CAACTTCATGATGGCCCGAG	
*Sdc3*	CCTCCACGACAACACCATTG	147
	ATGAGCAGTGTGACCAGGAA	
*Vgf*	GACCATCGCTCGTACTCCAG	167
	AACAGAAGAGGACGGATGCC	
*Cpe*	CCAACGCAACCATCTCCG	101
	TGTAAGTTTGTAGTTTCCAGGCG	
*G3bp2*	GTTCAATCCCAGCCACCAAG	160
	TGCCAACGAAAAGCTGATGA	
*Cpne5*	GTGGAGTCAGAGAGCACCTT	125
	CTCATGTAGTGCAGGGACGT	
*Agap2*	CAGGAATGGACTTTGAGCCG	188
	CGGATCAGGACCAGATGAGT	
*Madd*	ACTCTCAAACGTCTGGTGGA	136
	GGGCACTTGACTTCTCTTCC	
*Ikbkg*	AATCTGAGAGGTCGGGTTCC	121
	TGCACCATTTCACTCAACTGG	
*Dpp8*	CCATCAACAGAGCAGCAGTC	135
	TTCCCACAGTTCCAATTCGC	
*Slc25a1*	CACGTTGGACTGTGCCTTG	133
	GCAGCTTCACCACTTCATCG	
*Shmt2*	CCTGGGGTCCTGTCTGAAC	141
	AGGGTCCAGATCAAAGGCTT	
*Dld*	AAAACATCCTTATAGCCACGGG	152
	TTCTACACCAATTACTCCTGCG	
*Ralb*	TCCGTGACAACTACTTCCGA	100
	GACTTGTTGCCCACGACC	
*Itch*	CTGCATCGAGAAAGTTGGGA	84
	CTCTTGTAGGGTGGGAGGTC	
*Fyn*	ACTCTTCCTCTCACACTGGC	104
	ACTCAGGTCATCTTCCGTCC	
*Slc1a2*	CACCGTTGCCAGATCGTG	117
	CTGAGGTGGCTGTCGTGC	
*Slc1a3*	GGCAAGCCCTGAGAGATTTC	107
	CCCTGCGATCAAGAAGAGGA	
*Akt1*	CAAGGAGATCATGCAGCACC	112
	CCTGGTATCCGTCTCAGAGG	
*Cyfip2*	CATCTGTGTGGATTACTACGAGA	81
	CCAAAGCCCATCACCTTGAG	
*S100b*	GAGGACACCAGCAGCAAAC	125
	ACCCTCTCGTCCTGAATACTG	
*β-act*	AGCCATGTACGTAGCCATCC	115
	ACCCTCATAGATGGGCACAG	

## Data Availability

Data is contained within the article and Appendix A.

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
