# Peer review of "Changes in the Proteomic Profile After Audiogenic Kindling in the Inferior Colliculus of the GASH/Sal Model of Epilepsy"

_ijms, 2025, doi:10.3390/ijms26052331_

Round 1

Reviewer 1 Report

Comments and Suggestions for Authors

The authors examined molecular and immune changes in the inferior colliculus of GASH/Sal hamsters, a model for audiogenic epilepsy, by comparing non-responder animals (those with reduced seizure severity after repeated stimulation) to naïve hamsters. They used auditory brainstem responses, cytokine profiling, and proteomic analysis to identify key inflammatory markers and differentially expressed proteins linked to seizure susceptibility. Bioinformatics approaches revealed disrupted metabolic and inflammatory pathways, and gene expression validation supported these findings.

There are a few concerns I would like the authors to address:

First, the classification of animals into "responders" and "non-responders" based on the severity index lacks clarity. It would be helpful to specify if an objective statistical criterion, such as cluster analysis or another quantitative method, was used to determine the cutoff between these groups. If not, this needs to be discussed in the manuscript.

Additionally, the manuscript mentions using a modified scale for audiogenic seizures, but does not explain how it was adapted or validated. A brief explanation would clarify this.

The authors also observe a reduction in wave I amplitude in auditory brainstem responses (ABR), which they attribute to decreased auditory nerve activity. However, alternative explanations, such as changes in neuronal excitability or synaptic transmission, are not discussed. A more detailed exploration of these possibilities would improve the manuscript.

Furthermore, the relationship between ABR changes and seizure severity is not explored. Given the study's focus on seizure susceptibility, examining whether wave I changes correlate with seizure severity would provide valuable insights.

The cytokine analysis compares non-responders to naïve animals but excludes responder animals. Including responders in the analysis could offer deeper insights into the immune mechanisms that distinguish seizure resistance from susceptibility. This is true for all the following experiments that included naïve animals and non-responders. Please add a discussion of the choice.

Concerning the proteomic analysis, I have a few additional points: The handling of missing values (NA) is not fully explained. The authors mention eliminating 921 proteins, but it is unclear why these proteins were excluded and what method was used to handle missing data. Providing more detail on this process would help the reproducibility of the analysis.

The authors also report different peptide and protein numbers in the first and second experiments, but they do not clarify how this discrepancy was addressed, such as through normalization. This should be discussed to ensure that the interpretation of the results is not affected.

The description of the UMAP analysis is also vague. The authors mention that "even though analyzing all the proteins doesn’t result in perfect separation into distinct clusters," there are areas where samples cluster together. It would be more helpful to specify whether these clusters are statistically significant and how they should be interpreted in the context of the study.

Gene set enrichment analysis (GSEA) is mentioned, but the results and how they support the conclusions about epileptic processes are not sufficiently explained. More details about the outcomes of GSEA would strengthen this section of the manuscript.

Additionally, no correction for multiple comparisons is applied in the univariate t-test analysis, which identifies 159 DEPs. Given the large number of proteins analyzed, corrections for multiple testing (e.g., Bonferroni or FDR) are necessary to reduce the risk of false positives. This should be addressed in the analysis.

Finally, the results for Slc1a2, Akt1, and S100b in the WGCNA-selected candidates seem contradictory. While Slc1a2 and Akt1 show increased expression in the non-responder group, suggesting a response to neuronal stress or excitotoxicity, S100b shows reduced expression, which is unexpected since it typically increases during neuroinflammation or after seizures. The increased expression of Slc1a2 and Akt1 may indicate a compensatory response to epileptic activity, while the decreased expression of S100b suggests lower glial activation. This contradiction points to a potentially complex and dysregulated response in the non-responders and warrants further discussion to clarify the role of these genes in seizure resistance or neuroprotective mechanisms.

Minor:

Supplementary Figure 1 contains text that is too small, making it difficult to read and follow.

Reviewer 2 Report

Comments and Suggestions for Authors

This study investigates the proteomic and immune responses triggered by audiogenic kindling in the inferior colliculus, comparing non-responder animals, which exhibit reduced seizure severity following repeated stimulation, to GASH/Sal naïve hamsters. Bioinformatic analyses, including Gene Set Enrichment Analysis (GSEA) and Weighted Gene Co-expression Network Analysis (WGCNA), revealed disrupted pathways associated with metabolic and inflammatory epileptic processes and identified a module potentially linked to an increased seizure threshold. Additionally, differential gene expression analysis confirmed the upregulation of Gpc1, Sdc3, Vgf, Cpne5, Agap2, and Dpp8 and the downregulation of Ralb and S100b, aligning with reduced seizure severity. The authors conclude that these findings may uncover key proteomic and immune mechanisms underlying seizure susceptibility, offering potential therapeutic targets for refractory epilepsy.

Main Criticisms

Overall, the article is well-written and supported by data; however, several aspects require further clarification and refinement:

  1. Structure and Readability
    • The manuscript is difficult to follow due to its structure, which jumps directly from the Introduction to the Results, requiring the reader to infer key methodological details from the Abstract.
    • The Materials and Methods section is placed at the end, before the Conclusions, making it challenging for readers to understand experimental procedures before encountering the results. It is recommended to position the Materials and Methods section earlier in the manuscript.
  2. Abstract
    • The abstract contains multiple abbreviations without their full definitions. These should be clearly explained upon first use.
  3. Introduction
    • The authors should include a hypothesis or prediction based on the current state of research
    • The term GASH/Sal is defined in the abstract but not when it first appears in the body of the text. This should be corrected for clarity.
  4. Materials and Methods
    • In Figure 4 (ABR waves), no significant differences were observed between the left ear (blue) and the right ear (red), nor between the initial (t0) and final time points of audiogenic kindling. This suggests that these waves do not influence seizure susceptibility. The authors should discuss the implications of this finding.
    • Line 663 states that "we used the Genetic Audiogenic Seizure Hamster (GASH/Sal) provided by the Experimental Animal Service of the University of Salamanca," but it does not mention the number of animals per group (n). Additionally, the sex of the animals is not immediately specified; this information appears later in subsequent paragraphs. It should be presented upfront for clarity.
  5. Statistical Analysis
    • It is not explicitly stated which dependent variable was measured in the t-test for each protein (e.g., protein abundance). Currently, the reader must infer this from the Results, since the Methods appear later in the text.
    • The t-test is a parametric test, which assumes normal distribution and homogeneous variances between groups. The authors must justify these assumptions before using this statistical approach.
    • The effect size is typically a value between 0 and 1, where 0 indicates no effect and 1 represents a total effect. Since effect size influences sample size calculations, the authors should discuss whether their findings align with the a priori sample size estimation of the study.
  6. Figures
    • Some figures require higher resolution to improve readability and data interpretation.

Reviewer 3 Report

Comments and Suggestions for Authors

This is an interesting study by Zeballos et al. investigating the key proteomic and immune mechanisms underlying epileptogenesis using a genetic model for audiogenic epilepsy in hamsters (GASH/Sal). The authors report an upregulation/downregulation of a number of genes related to metabolic and inflammatory processes in inferior colliculus, which might be a finding relevant to epilepsy research. Overall, the study is organized in a concise manner and conclusions are largely supported by the reported results. However, there are some minor and major concerns that need to be addressed:

  1. The seizure severity gradually decreased during the course of the stimulation protocol, which is opposite to other kindling procedures in the epilepsy field. This phenomenon needs to be explained in detail. Also, do GASH/Sal animals have spontaneous seizures?
  2. The authors first make a valid argument that the two distinct populations emerged in their sAUK protocol (responders and non-responders) but then none of the follow-up experiments include this comparison. For example, does the significant decrease in amplitude of the first wave in sAUK.NR animals can account for differences in seizure severity?
  3. It is not fully clear why the authors chose to only compare sAUK.NR and naïve animals. The rationale for this decision needs to be thoroughly explained. If the data from the responder group is available, it should be included in the manuscript.
  4. No information is provided on how the sample size was determined. The exact P-values should be reported.
  5. There’s a typo in the Y-axis on Figures 1 and 2 (“Seveity index”).

Comments on the Quality of English Language

Although the quality of English usage is rather high throughout the manuscript, there's room for improvement, particularly in the Discussion section.

Round 2

Reviewer 1 Report

Comments and Suggestions for Authors

I thank the authors for carefully addressing all my comments and for providing thoughtful answers to my concerns. I think the manuscript improved in clarity and congratulate the authors on their efforts.

Minor: I understand your point for supplementary figure 1. Doesn't matter as far as the dpi are at 900. Thanks.

Reviewer 3 Report

Comments and Suggestions for Authors

Although the authors have provided adequate responses to my concerns, I still think that including the responder group in at east some analysis would have been interesting and relevant (to serve as a kind of "positive control"), especially since hamsters in the Naive group actually never experienced a single seizure, thereby serving as a true "negative control".